# Competition and cooperation: The plasticity of bacterial interactions across environments

Josephine Solowiej-Wedderburn [1,2], Jennifer T. Pentz[3], Ludvig Lizana[2,4], Bjoern O. Schroeder[5,6,7], Peter A. Lind[5,7], Eric Libby[1,2,7]*

**1** Department of Mathematics and Mathematical Statistics, Umeå University, Umeå, Sweden, **2** Integrated Science Lab (IceLab), Umeå University, Umeå, Sweden, **3** Los Alamos National Laboratory, Los Alamos, New Mexico, United States of America, **4** Department of Physics, Umeå University, Umeå, Sweden, **5** Department of Molecular Biology, Umeå University, Umeå, Sweden, **6** Laboratory for Molecular Infection Medicine Sweden (MIMS), Umeå University, Umeå, Sweden, **7** Umeå Center for Microbial Research (UCMR), Umeå University, Umeå, Sweden

\* elibbyscience@gmail.com

**Data availability statement:** There are no primary data in the paper. All code used is

## Abstract

Bacteria live in diverse communities, forming complex networks of interacting species. A central question in bacterial ecology is whether species engage in cooperative or competitive interactions. But this question often neglects the role of the environment. Here, we use genome-scale metabolic networks from two different open-access collections (AGORA and CarveMe) to assess pairwise interactions of different microbes in varying environmental conditions (provision of different environmental compounds). By computationally simulating thousands of environments for 10,000 pairs of bacteria from each collection, we found that most pairs were able to both compete and cooperate depending on the availability of environmental resources. This modeling approach allowed us to determine commonalities between environments that could facilitate the potential for cooperation or competition between a pair of species. Namely, cooperative interactions, especially obligate, were most common in less diverse environments. Further, as compounds were removed from the environment, we found interactions tended to degrade towards obligacy. However, we also found that on average at least one compound could be removed from an environment to switch the interaction from competition to facultative cooperation or vice versa. Together our approach indicates a high degree of plasticity in microbial interactions in response to the availability of environmental resources.

## Author summary

Bacteria live in diverse communities, with a range of different interactions between species. An open question in microbial ecology is whether competitive or cooperative interactions are more prevalent. While this question has been studied in specific

available on a GitHub repository: https://github.com/josephine-solowiej-wedderburn/CompCoopEnvPaper.git and deposited on Zenodo: https://doi.org/10.5281/zenodo.15075550.

**Funding:** JSW acknowledges funding from grant JCK-2129.2 from Kempestiftelserna (https://www.kempe.com) that paid her salary. BOS acknowledges funding grant 2021-06602 from the Swedish Research Council Vetenskapsrådet (https://www.vr.se). The funders had no role in study design, data collection and analysis, decision to publish, or preparation of the manuscript.

**Competing interests:** The authors have declared that no competing interests exist.

environmental contexts, the effects of changing environments are often neglected. To address this gap, we employed a data-driven, theoretical approach to investigate how environmental resources shape bacterial interactions. We used genome-scale metabolic networks from two different open-access collections (AGORA and CarveMe) to predict how different pairs of bacteria interact under a variety of environmental conditions. Scanning thousands of environments for thousands of pairs of bacteria, we found that the vast majority of bacteria can both compete and cooperate depending on their environment. We also discovered that cooperation is more common in resource-poor environments. Our analysis of dynamic environments revealed frequent shifts between cooperative and competitive interactions, showing how changing environmental conditions can shift bacterial relationships. Together these findings indicate that microbial interactions are highly plastic to the availability of environmental resources, highlighting the complexity and adaptability of microbial life.

## Introduction

A core question in ecology is the nature of the interaction between two organisms. For microorganisms such as bacteria this question can be challenging since organisms often interact indirectly by modifying their chemical environment. If two organisms consume the same resources then there can be competition which in simplified systems can drive one species extinct, e.g., competitive exclusion [1]. Yet when two organisms secrete resources needed by the other there can be cooperation, e.g., cross-feeding [2,3]. Within these two extremes there is a large range of interactions and predicting which one actually occurs has important consequences for microbial eco-evolution and our ability to design microbial communities that perform useful functions [4–8]. Despite the significant interest in this topic there are still many fundamental questions that remain.

One fundamental question is whether a random pair of microbes is more likely to compete or cooperate. Recent research has increasingly pointed toward competition as the predominant interaction [9]. However, investigations into both natural and synthetic microbial communities have also revealed an abundance of cooperative interactions [3,10,11]. Even within the same context, such as the human gut, studies can differ on which interaction dominates whether it is competition [12–14] or cooperation [15–17]. Adding to the complexity, studies have indicated that environmental factors such as temperature, toxicity, and resource availability can all contribute to the interactions between pairs of microbes and that sometimes the same microbes can exhibit vastly different types of interactions under different conditions [18–21]. Given the many possible sources of competition and cooperation, it can be difficult to resolve the discordance between these studies. For clarity, we focus on environmental resources as an axis along which interactions may vary. In this context, competition arises via limited resources and cooperation arises via the production of new resources or the removal of waste products [2,22–26].

Even by narrowing our focus to the role of environmental resources in shaping interactions, there is considerable complexity. For example, environments may select for specific characteristics in species which can lead to distinct interaction patterns. In host-associated environments, microbial species tend to have smaller genomes and may rely on other species for particular resources, whereas free-living habitats are often occupied by species with larger genomes and overlapping nutritional requirements [16]. These broad scale generalizations fail to capture many scenarios and often overlook salient details. For instance, the same set of microbes can interact differently following relatively minor changes to environmental

resources. In cultured microbes that cross-feed via amino acids, experiments adjusting the availability of as little as two resources can lead to different interactions [27]. Similarly, in the human gut dietary changes can have significant impacts on communities of microbes. Normally a variety of dietary fibers maintains a complex and diverse microbial community, but restricted fiber consumption can lead to the decline of fiber-restricted specialists in place of generalists, that then switch to the mucus barrier of the host as an alternative energy source [28–30]. Taken together, these findings suggest that the environment plays a complex and pivotal role in shaping microbial interactions.

Despite the number and complexity of possible interactions between microbes, experiments studying how microbial communities assemble have found highly reproducible community structures [31–33]. These reproducible community structures indicate the presence of general mechanisms that may drive cooperative and competitive interactions between members of a community. By comparing the metabolic requirements and capabilities of species, studies have found predictive patterns. Cooperation more often emerges between metabolically dissimilar species [34,35] and can contribute to establishing diverse communities [36, 37]. Conversely, competitive interactions more often occur between metabolically similar species due to competition for available resources and metabolic niche overlap. This has been shown to drive predictable community assembly and structure in both well-controlled laboratory environments [31,38] as well as more complex environments such as within the gut of the nematode *Caenorhabditis elegans* [14] and on the surface of an *Arabidopsis thaliana* leaf [39]. While these studies have shed light on the patterns governing community assembly, it is unclear the extent to which they might be observed outside of controlled lab settings.

We can address the general nature of interactions by adapting tools used to study the microbiome and community assembly. Since a common focus in both has been on the role of metabolic interactions, genome-scale metabolic models provide a useful tool [40,41]. These models contain a set of chemical reactions inferred from genome annotations and can be used to predict growth rates given a set of environmental conditions (usually in terms of supplied chemical compounds). In recent years, metabolic models have been curated for thousands of species with a wide range of applications from the development of personalized medicine [42] to exploring the rarity of prokaryotic endosymbioses [43]. Such models have also been used to identify features of different metabolisms that lead to competition or cooperation between species [16,44–46]. In the context of the human gut, these models have been used to explore the effects of different diets on interactions between the microbiota [17,47] and, more broadly, identify environments which promote, e.g., obligate interactions between bacteria [48]. However, metabolic modeling has typically been used to examine specific sets of organisms or specific environmental contexts. By harnessing the computational tractability of these metabolic models and leveraging the huge number of models now available, we can interrogate the role the environment plays in shaping competition and cooperation more broadly, on a scale that would be difficult to evaluate empirically.

Here, we use metabolic models from two different sources (AGORA [17] and CarveMe [49]) to assess the interactions between thousands of random pairs of bacteria across thousands of different environments. We categorize interactions in terms of their effects on growth by using flux balance analysis to compare the growth rate of metabolisms in pairs and separately in different environments. In this context, cooperation is when both metabolisms grow faster in each other's presence and competition requires that at least one grows slower (i.e. there must be at least one "loser"). We vary the composition and the number of compounds in environments to elucidate general principles concerning how environments affect the likelihood of competition and cooperation. By systematically removing compounds from the environment we measure the robustness of interactions to environmental fluctuations and how interactions

change as environments degrade. Ultimately, we find that interactions between bacteria show a high degree of plasticity depending on the environment, with a systematic trend towards obligate interactions in less diverse environments.

## Results

### Classifying interactions

We assessed how the environment can shape ecological interactions by using metabolic models from two of the largest open-access collections of genome-scale metabolic networks available: AGORA [17] and CarveMe [49]. The AGORA collection includes models for 818 strains of bacteria from the human gut, while CarveMe has models for 5,587 strains of bacteria from disparate places, using genomes from NCBI RefSeq (release 84, [50]). Since the species in AGORA are from the same environment, in contrast to those in CarveMe, we can compare the interactions between bacteria that commonly co-occur relative to those that may never have interacted. In both collections, metabolic models partition metabolites into intracellular and extracellular compartments. We refer to the availability of extracellular compounds as the 'environment'. By modifying the presence and abundance of these compounds, (e.g., amino acids, carbohydrates, or metal ions), we can assess the role of the environment on bacterial growth, as computed by the metabolic models.

In both collections, every model comes with a default environment: a set of compounds and concentrations that ensures the growth of the corresponding organism. To classify the potential interaction between a bacterial pair, we compared the growth rates of the organisms together versus alone in a new joint environment. The joint environment was created by combining the default environments of the two bacteria such that both were guaranteed to grow. We drew a distinction between two qualitatively different interactions: competitive and cooperative (see Fig 1A). Competitive interactions are identified when at least one of the organisms must grow slower when both are present. Cooperative interactions are identified when it is possible for both organisms to grow faster when grown together (see *Methods: Assessing growth of individuals and interactions of pairs*). Other interactions are also possible, including neutral interactions which arise when the growth rate of each organism is unaffected by the presence of the other. We considered 10,000 random pairs from each collection and classified their ecological interaction using their joint default environments. We found that the most common default interaction in both collections is neutral (see Fig 1B), 49% in AGORA and 59% in CarveMe. Of the remaining interactions, we found that the CarveMe collection has more competitive interactions than AGORA and both collections have few cases of cooperation (2% AGORA and 0% CarveMe).

### Environment-driven variation in competition and cooperation

To determine whether the interactions present in the default environments were definitive or plastic, we took the same sets of 10,000 pairs of bacteria (from Fig 1B) and evaluated their interactions in different random environments. We constructed these environments by providing the essential compounds that the bacterial pair must have in any environment in order to grow with no possible substitutes (see *Methods: Essential compounds for pairs*, S1 and S2 Figs, and S1 Table). In addition to the essential compounds, we randomly added a fixed number of environmental compounds that could be utilized by at least one of the bacteria (50 additional compounds for smaller environments; or 100 for larger, see *Methods: Algorithm for generating environments*). For simplicity, we set the availability of all environmental compounds to a fixed but limited amount that is sufficient for growth but can induce competition

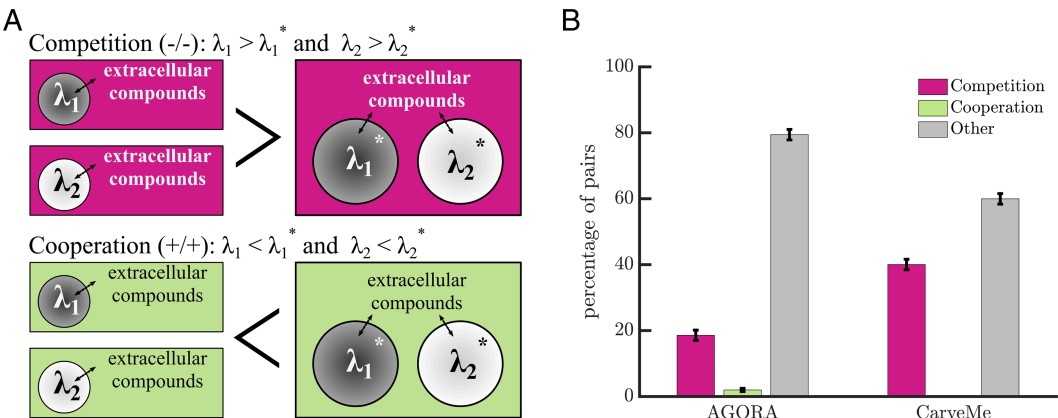

**Fig 1. Classifying interactions in default environments.** A) A schematic shows interactions between pairs of bacteria (circles) in different environments (rectangles). Competition or cooperation are determined by comparing two computed growth rates for each species in a given environment: 1) the maximal growth rate of each bacteria in isolation ($\lambda_1$ and $\lambda_2$); with 2) the maximal growth rate of each bacteria without harming their partner when grown together ($\lambda_1^*$ and $\lambda_2^*$, see *Methods: Assessing growth of individuals and interactions of pairs*). We note that with this framework, the same pair of bacteria could have different interactions depending on the environment. B) A bar graph shows the percentage of interactions between 10,000 random pairs of bacteria in their default environments from AGORA and CarveMe. The percentages are the means of 10 samples of 1,000 random pairs; an ANOVA confirms that percentages do not differ significantly across bins (*p*-value > 10%). In both collections cooperation is rare in comparison to competition. Over half of the observations were 'other' and of these interactions, neutral was most common; the remaining other interactions were commensal (+/=).

should both organisms use it (see *Methods: Algorithm for generating environments*). Since the most common interaction in the default environments was neutral, we looked for environments that produced competition or cooperation. For each pair of organisms, we sampled each type of environment until we found 100 viable growth environments. Across all samples, we found at least one cooperative and one competitive environment for 67% (standard deviation 1.2%) of pairs in AGORA and over 87% (standard deviation 0.75%) of pairs in CarveMe (Fig 2A).

While the majority of bacterial pairs have the metabolic capacity to both compete and cooperate, the likelihood of finding environments for these interactions differs. Fig 2C shows the proportion of environments with competitive, cooperative, or other interactions for different pairs. We observed more examples of cooperative environments than competitive environments across both collections. Consistent with the data from default environments, we found that a higher proportion of environments were competitive in CarveMe in comparison to AGORA. We also found that the relative frequency of observing competitive or cooperative interactions changed as we varied the number of environmental compounds. For example, cooperative interactions were most frequent in the low diversity (small) environments. Conversely, in high diversity (large) environments, there was a more varied spread of interactions.

Fig 2C indicates that there is variability in the number of competitive or cooperative interactions found between different pairs. Since the underlying organisms represent a wide spectrum of metabolic complexity (e.g., the number and types of reactions present in their metabolic networks), it is possible that some measure of their different metabolic capacities may be predictive of their tendency towards competitive or cooperative interactions. We investigated this possibility by quantifying metabolic complexity in two ways: 1) the number of environmental compounds that a metabolism could use, or 2) the number of metabolic

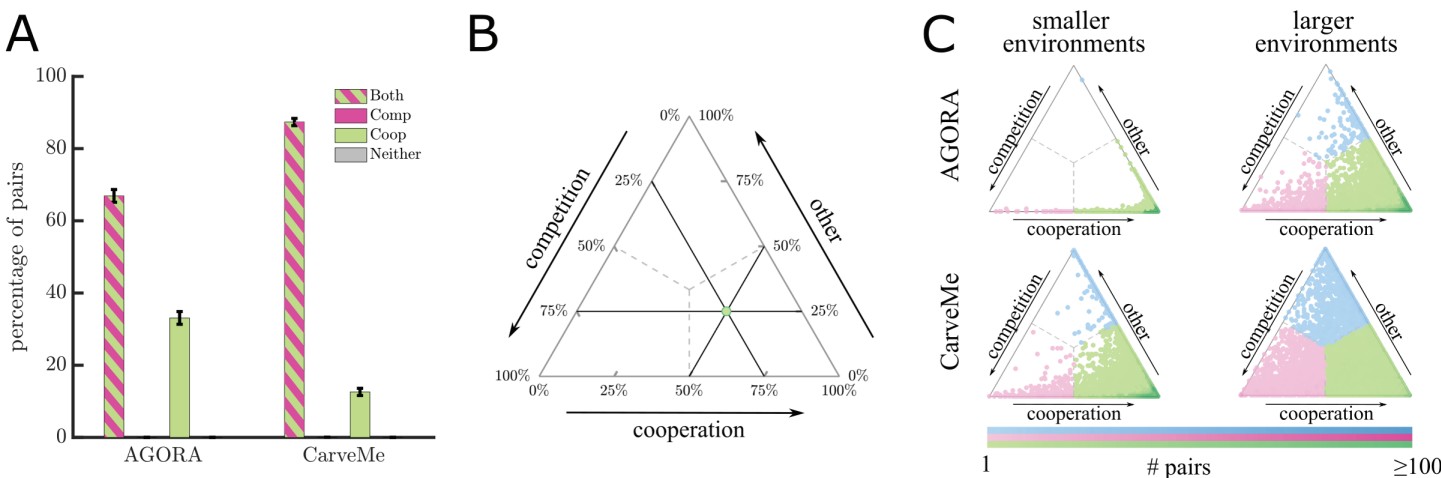

**Fig 2. Environment-driven variation in competition and cooperation.** A) Bar chart shows the proportions of bacteria pairs where it was possible to find at least one environment for potential cooperation and/or competition in AGORA and CarveMe. Striped bars indicate pairs where we found both competitive and cooperative environments; solid colors indicate just competitive (pink) or cooperative (green) environments were found. The percentages are the means of 10 samples of 1,000 random pairs, as in Fig 1B; an ANOVA confirms that percentages do not differ significantly across bins (*p*-value > 10%). At least one cooperative and one competitive environment were found for 67% (standard deviation 1.2%) pairs in AGORA and 87% pairs (standard deviation 0.8%) in CarveMe; and at least one cooperative environment was found for over 99% pairs across both collections. B) Schematic for triangle plot showing the proportion of different interactions found per pair of bacteria. The three sides of the triangle represent the proportion of competitive, cooperative, or other (neutral or commensal) environments identified, so the example point represents a pair of bacteria where we found competition in 25% viable environments sampled, cooperation in 50% viable environments, and other interactions in the remaining 25%; as we found more cooperative environments than competitive or other, the point is shaded green. C) Triangle plots show the proportion of different interactions across the 100 viable growth environments identified for pairs of bacteria from (A), as explained in (B). Top row shows results for AGORA and bottom row for CarveMe. Smaller environments (essential compounds plus 50 compounds) are shown on the left and larger environments (essential compounds plus 100 compounds) are shown on the right. The color of points represents their closest vertex and the darkness indicates the density of points. All four plots have the highest density of points in the bottom right corner, shaded in dark green, indicating a high density of pairs (≥ 100) where we only found environments for cooperative interactions. A permutation test of the proportions of interactions found in smaller and larger environments confirms that there is a statistically significant difference between the two environment sizes (*p*-value <0.001 for 10,000 permutations in both AGORA and CarveMe). We found a more diverse spread of interactions in larger environments.

reactions it could perform. For each measure of metabolic complexity, we separated species into three categories based on percentiles: low, medium, and high (see S3A, S3B, S4A, and S4B Figs). In all cases, we rejected the null hypotheses that the number of competitive or cooperative cases was uniform across the different pairings of categories (*p-value* <0.001, S3C–S3F and S4C–S4F Figs). We consistently found the fewest cooperative environments between low–low metabolic complexity pairings. We observed other differences that were statistically significant in particular comparisons, but they were not significant across all comparisons, taking into account model collections, measures of complexity, or types of pairings (e.g., high-low or medium-high). Thus, these analyses demonstrate that although there are differences in interactions owing to metabolic complexity, they are not consistent enough to indicate a general rule.

If we delve deeper into our data of cooperative and competitive interactions, we observe that there are different types of cooperative interactions. We organized these into three categories: 1) facultative where both microbes can survive alone but have the potential to grow faster together ($+/+$); 2) one-way obligate where one microbe is reliant on the other and the second microbe can survive alone but benefits from the pairing ($\times/+$); 3) two-way obligate where neither microbe can grow without the other ($\times/\times$). Fig 3 shows that the most common form of cooperation found in our random sampling of different environments was one-way obligate. In ≈76% cases (78.3% AGORA and 72.9% CarveMe) the bacteria that was able to survive alone in one-way obligate interactions had a higher metabolic complexity (both when

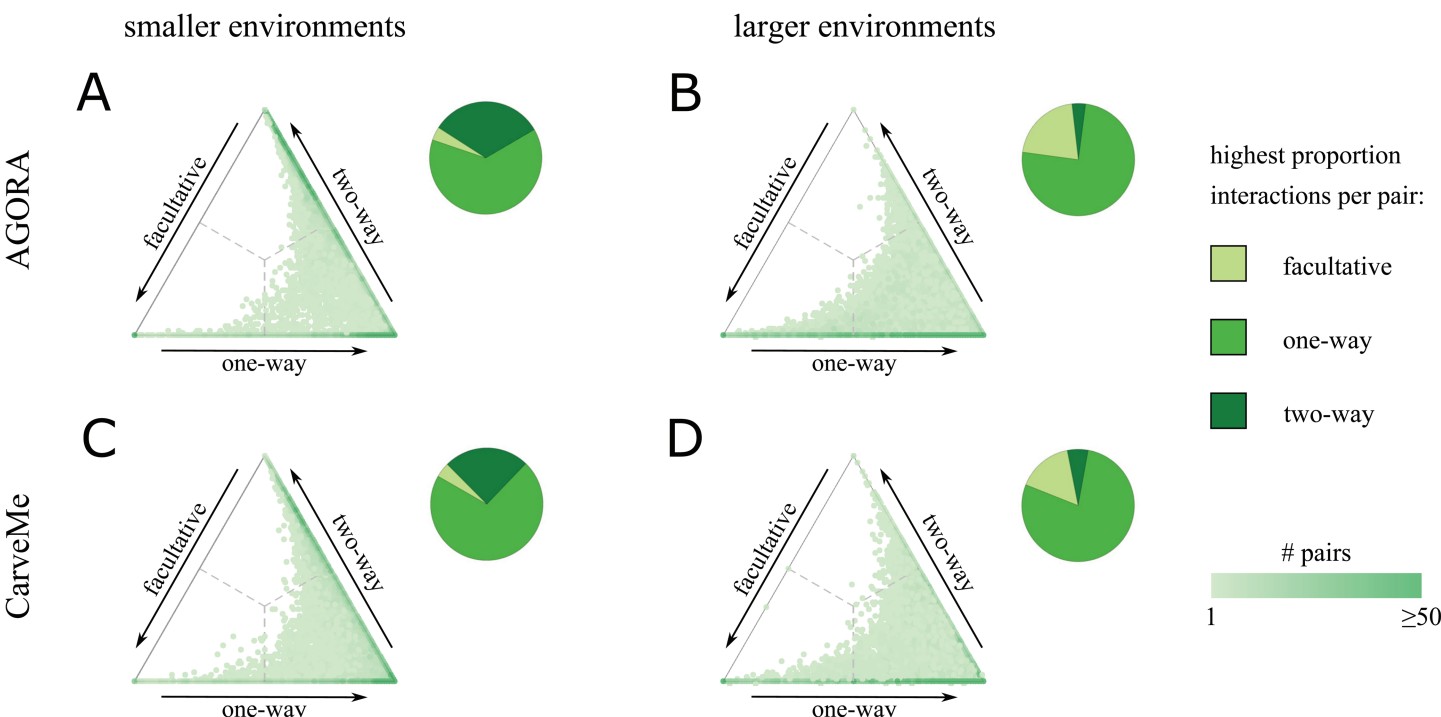

**Fig 3. Environment-driven variation in types of cooperation.** A–D) Triangle plots show the proportion of different types of cooperative environment: facultative, one-way obligate or two-way obligate, corresponding to pairs of bacteria in Fig 2C. Smaller environments (essential compounds plus 50 compounds) are shown in the left panel (A and C) and larger environments (essential compounds plus 100 compounds) are shown in the right panel (B and D). Top row (A and B) shows results for AGORA and bottom row (C and D) for CarveMe. The three sides of the triangle represent the proportion of environments for each of the three types of cooperative interaction identified, in the same way that competition, cooperation and other environments are represented in Fig 2B. Each point on the triangle plots corresponds to a pair of bacteria. Points are shaded by density such that darker shades of green correspond to a higher density of pairs. Pie chart inserts show the fraction of pairs for which the highest proportion of cooperative environments identified were facultative (light green), one-way obligate (green) or two-way obligate (dark green). One-way obligate cooperation was the most common interaction found in both collections and environment sizes. However, a permutation test of the proportions of cooperative interactions found in smaller and larger environments confirms that there is a statistically significant difference between the two environment sizes ($p$-value <0.001 after 10,000 permutations in both AGORA and CarveMe). We found more cases of two-way obligate interactions in smaller environments and more cases of facultative cooperation in larger environments.

measured by capacity to use environmental compounds or number of reactions). In environments with a high resource diversity (100 additional compounds), the next most frequent type of cooperation was facultative. This differed from environments with a low resource diversity (50 additional compounds) where two-way obligate interactions were the next most frequent type of cooperation observed.

## Stability of interactions to the loss of a single compound

Given that specific environments facilitate either competitive or cooperative interactions between bacteria, our next step was to investigate how robust these interactions are to perturbations of the environment. We investigated what would happen if we randomly removed a single compound from competitive or cooperative environments for 1,000 random pairs in each collection. In the case of competitive environments, Fig 4A and 4C show that the interactions are robust to the loss of single compounds; however, we find that approximately 3–6 of the 100 environmental compounds (means of 6.5 compounds from AGORA and 2.8 from CarveMe) can switch the competitive interaction to cooperative (facultative or obligate). In

the case of cooperative starting environments, we exclusively considered facultative cooperation because obligate interactions cannot lose their obligacy when compounds are removed. We found that, similar to competitive environments, cooperative interactions are robust to the loss of a single compound, though approximately 1 of the 100 compounds (means of 1.0 compounds from AGORA and 1.4 from CarveMe, Fig 4B and 4D) caused a switch to competition. There are additionally an average of 2-3 compounds (means of 3.5 compounds from AGORA and 2.1 from CarveMe) that cause a transition from facultative to obligate cooperation when removed.

The previous analyses demonstrated that the difference between competitive and cooperative interactions often depends on a single compound in the environment. Based on this observation, we sought to identify whether any compounds were hallmarks of competitive or cooperative environments. We compared the frequency of compounds in the various environments and found that all compounds appeared in both competitive and cooperative environments (see S5A and S5B Fig). We did find that the majority of compounds (> 95%) appeared in a higher proportion of competitive or cooperative environments than by independent random sampling ($p$-$value$<0.001), though the bias was small (the median bias was only a difference of 12.1% in AGORA and 8.0% in CarveMe, see S5C and S5D Fig). As a case study, we took two groups of environmental compounds that have previously been identified as playing key roles in microbial interactions—amino acids [2,3] and simple sugars (aldoses and ketoses) [37,51]—in S6 and S7 Figs. While we found there was a statistically significant difference in the mean number of amino acids or simple sugars between competitive or cooperative environments, the difference was of the order of 1 compound or less. For example, in CarveMe we would expect to find 9 amino acids in both competitive and cooperative environments (means 9.3 and 8.9, respectively), while in AGORA we found an average of one more amino acid in competitive than cooperative environments (means 14.1 and 13.1, respectively).

Instead of looking for single compounds as hallmarks for environments, we shifted to consider broader measures of environmental differences. We used the Jaccard distance as a metric to determine the difference/similarity between competitive and cooperative environments. If two environments are identical they have a Jaccard distance of 0, while if they are completely distinct they have a distance of 1. We first applied this metric within competitive and cooperative environments and found a diverse spread of environments for both (mean distance 0.58–0.82, S8A–S8D Fig). We then calculated the Jaccard distance between competitive and cooperative environments and found similar distances (means 0.60–0.81), indicating that it is difficult to distinguish between competitive and cooperative environments with a simple similarity metric, even on a pair-by-pair basis (S8E–S8H Fig).

If we shift our focus to those compounds responsible for switches between competitive and cooperative interactions, we found 45–58% of environmental compounds were responsible for all the interaction switches between competitive and cooperative environments in Fig 4A–4D (45% in AGORA and 58% in CarveMe; results shown in S9 and S10 Figs). Here, the majority of compounds were responsible for fewer than 1% of switches, however, the most frequently observed compounds to cause a switch were in 5–12% of the respective interaction switches, which is statistically significant. Supplemental S2 and S3 Tables list the compounds which most commonly caused switches between interactions when removed from the environment. This includes key compounds in energy metabolism: terminal electron acceptors for aerobic (oxygen) and anaerobic (nitrate, nitrite, fumarate) respiration. There is an overlap in compounds that would cause a switch from competitive to cooperative interactions with those that would cause a switch from cooperative to competitive interactions. For example, removing oxygen from the environment commonly caused interaction switches from both competition to cooperation and cooperation to competition between pairs in AGORA. In particular,

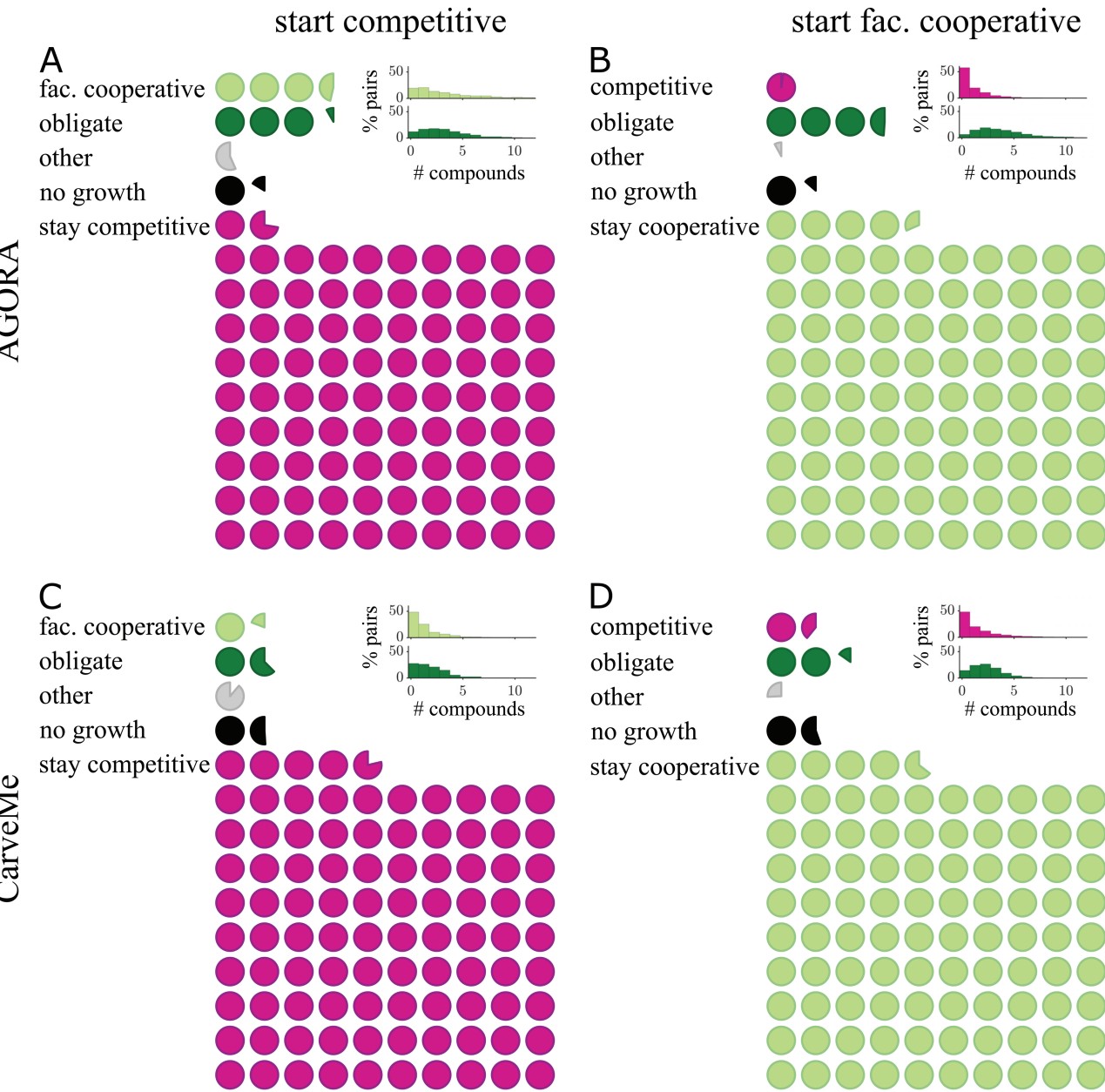

**Fig 4. Stability of interactions to the loss of a single compound.** A–D) Circles summarize the average effect on the interaction between a pair of bacteria when a single compound is removed from the environment. In (A) and (C) the initial environment is competitive, and in (B) and (D) facultative cooperative. Specifically, 1,000 pairs of bacteria in AGORA (A and B) or CarveMe (C and D) are considered; for each pair we select one competitive and one facultative cooperative initial environment with 100 additional compounds, determine the effect when each of these compounds is individually removed from the starting environment (see *Methods: Single removal of compounds*). The circles show the mean frequency that removing a single compound from the environment would result in a switch to a different type of interaction or stay the same (separating the data into 10 bins confirms there is no significant difference between bins, *p*-value > 10%). In the majority of cases, if a single compound is removed from the environment the interaction between the pair of bacteria would remain the same. However, on average there is at least one compound which would result in a switch from a competitive to a cooperative environment or vice versa. Qualitatively, the results for initial facultative cooperative environments appear to mirror the results when starting in competitive environments for both collections. Inserted histograms show that for 70-83% pairs there are 1–6 compounds which can cause a switch to obligate interactions; while for 41–60% pairs there are 1–6 compounds which can cause an interaction switch between competition and facultative cooperation if individually removed.

several amino acids were found to cause all types of transitions for both collections, with most (15–18) responsible for a statistically significant proportion of switches to obligate interactions and 7–15 responsible for significant proportions of switches between competitive and facultative cooperative interactions ($p$-value<0.01, S11 Fig). Other categories were less universal, for example, B-vitamins (nicotinate (conjugate base of niacin, B3) and riboflavin (B2)) were only common for switches to obligate interactions in AGORA, while switches to obligacy in CarveMe were commonly caused by removal of metal ions ($Fe^{2+}$, $Fe^{3+}$ and $Cu^{2+}$). By contrast, the removal of simple sugars from the environment less frequently caused a change in the interaction between pairs of bacteria (S12 Fig).

## Plasticity of interactions to the loss of environmental compounds

We further explored the plasticity of interactions by investigating what happens when environments deteriorate. We did this by systematically removing compounds in competitive and facultative cooperative environments and recalculating the exponential growth rates to evaluate whether the interaction changed (see schematic in Fig 5A and *Methods: Degradation of environments*). To mitigate the effect of redundant compounds which would have minimal ecological impact, we identified the compounds being 'used' by at least one species on a pair-by-pair basis and reduced environments to contain only these compounds before performing our degradation analysis (see S1 Fig). Furthermore, we never removed the compounds that had previously been identified as essential across all environments for that pair. Fig 5B–5G show example simulations of pairings of *Prevotella copri* with three other bacteria commonly occurring in the human gut (see S13–S18 Figs for other pairings between common bacteria in gut, aquatic or soil communities). Each figure shows a set of random sequences of compounds being removed (each column represents a different order of removal), beginning in a particular choice of either a competitive (Fig 5B–5D) or cooperative (Fig 5E–5G) environment. These different pairings highlight some common trends while illustrating the variation caused by the order of compound removal. For example, in the majority of cases that began with a competitive interaction, there was a transition to cooperative interactions. For the choice of initial competitive environment between *Prevotella copri* and *Clostridium perfringens* (Fig 5C) the switches to cooperation occurred after very few compounds were removed, while in the other pairings more compounds were removed before the switches occurred. Whether starting in a competitive or cooperative environment, the simulations between *Prevotella copri* and *Bacteroides vulgatus* (Fig 5B and 5E) show a high frequency of transitions between competition and cooperation, often returning back to the initial interaction, but in the other pairings we typically only observed one or two interaction switches. Finally, we note that the last state before the pairs could no longer grow was often (82%) an obligate interaction. (See S1 Appendix for a discussion on the metabolic complexity and environmental context that underlies the potential mechanisms governing these switches.)

Inspired by these case studies, we considered a larger set of pairs of organisms to see whether the observed trends are consistent. We performed this search for 300 random pairs of bacteria from each collection and starting in either competitive or facultative cooperative environments. Across all these simulations we found that it was common for bacteria to switch to another interaction type before they stopped growing. In AGORA, the competitive environments changed interaction with the removal of fewer compounds than from a cooperative environment (Fig 6A); while in CarveMe on average, the same number of compounds could be removed from both competitive and cooperative environments before the interaction changed (Fig 6B). In Fig 6C–6F we summarize the transitions between interactions for 50 different sequences of compounds removed across the different pairings, until at least one

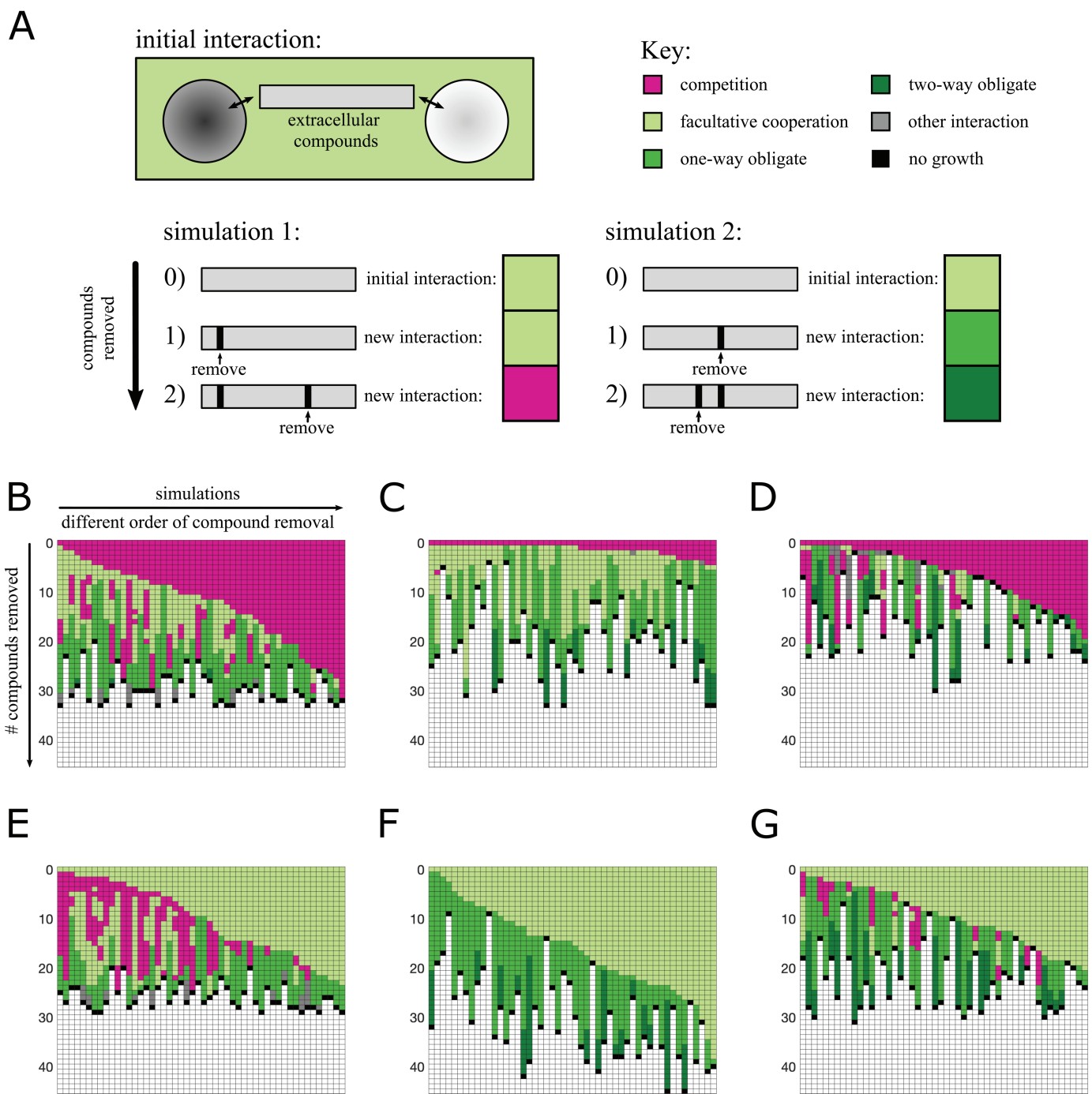

**Fig 5. Degrading competitive and cooperative environments for common gut bacteria.** A) A schematic shows how compounds are sequentially removed from an environment for facultative cooperation between a pair of bacteria and the resultant change in interaction. Colors correspond to different interactions: competition in pink; facultative cooperation in light green; one-way obligate interactions in green; two-way obligate interactions in dark green; no growth in black and other interactions in gray ( ( = / = ), ( +/ = ), or ( ✕/ = ) ). Two different simulations are illustrated where different compounds are removed from the same starting environment: in simulation 1 the interaction switches from facultative cooperation to competition after 2 compounds are removed from the environment, while in simulation 2 the interaction becomes one-way and then two-way obligate as compounds are removed. B–G) Heatmaps show how the interactions between three pairs of common gut bacteria change as compounds are removed from the environment (models from AGORA). Going down the columns subsequent compounds are removed from the environment. Each column shows the interaction shifts as different orderings of compounds are removed and columns have

been sorted by the number of compounds removed until the first change in interaction. In the top row (B–D) pairs begin in an environment for competition and in the bottom row (E–G) for facultative cooperation. Bacteria pairs *Prevotella copri DSM 18205* and *Bacteroides vulgatus ATCC 8482* (B and E), *Prevotella copri DSM 18205* and *Clostridium perfringens ATCC 13124* (C and F), and *Prevotella copri DSM 18205* and *Ruminococcus gnavus AGR2154* (D and G) show the variety of interaction changes that are commonly observed when environments are reduced. *Prevotella copri* appears to switch more frequently between competition and cooperation when paired with *Bacteroides vulgatus* than *Clostridium perfringens* or *Ruminococcus gnavus*. The most common final interaction before at least one bacteria cannot grow is obligate.

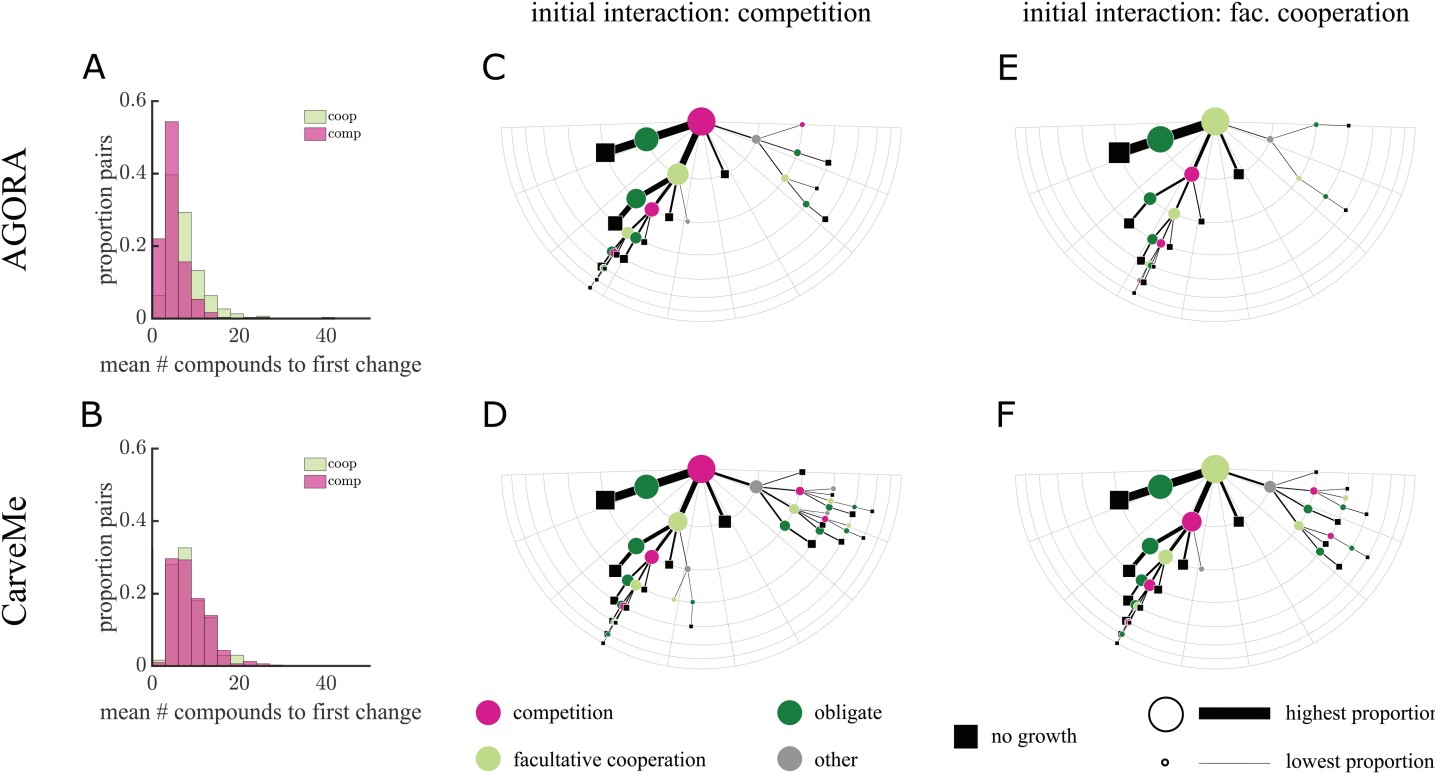

**Fig 6. Plasticity of interactions to the loss of environmental compounds.** A and B) Histograms show the number of compounds that can be removed before the first change in interaction type for 50 different sequences of compound removal for 300 pairs of bacteria, starting in either competitive or facultative cooperative environments. On average, competitive interactions in AGORA switch with the removal of fewer compounds than cooperative interactions (4.9 *vs.* 7.4; (A)), while competitive and cooperative interactions both switch after an average removal of ∼8 compounds in CarveMe (8.3 and 8.5; (B)). C–F) Diagrams summarize the change in interactions between 300 pairs of bacteria as compounds are removed from the environment. The left panels summarize interaction paths starting in competition ((C) AGORA and (D) CarveMe), and the right panels starting in facultative cooperation ((E) AGORA and (F) CarveMe). The semi-circular grid indicates the number of interaction switches. Nodes and line thickness are proportional to the square-rooted proportion of interactions taking that path; we only include paths that happened for > 0.1% simulations (see S19 Fig for full results). Additionally paths are sorted such that the most common branch at each node is furthest to the left. S20 Fig shows that the results here are significantly different to equivalent simulations with fixed probabilities of compounds causing given interaction switches (ANOVA *p*-value <0.001), showing the impact of environmental compounds on microbial interactions changes as environments degrade. In particular, we predict more obligate interactions as environments degrade than with fixed probabilities calculated in more diverse environments. Overall, the most common path whether starting in competition or facultative cooperation is to obligate interactions and then no growth.

of them was no longer able to grow. Regardless of the initial interaction, as compounds were removed the most common transition path moved to an obligate interaction and ultimately no growth (left-most path in Fig 6C–6F). The next most common path was to switch between competition and facultative cooperation once before transitioning to an obligate interaction and then ceasing growth. Interestingly, we also observed some sequences with frequent switches, e.g., ≥ 5 transitions between competition and facultative cooperation and vice versa, highlighting the complex relationship between environment and ecology.

## Discussion

A fundamental question in ecology is the nature of the interaction between two species. For bacteria these interactions are typically placed on a spectrum between competition and cooperation, with conflicting studies reporting which is more prevalent. While there is an appreciation of some plasticity in specific interactions based on the availability of environmental resources [27], it remains unclear to what extent this applies. Here, we assess the extent to which the environment can modulate ecological interactions by using a large-scale computational approach. By sampling thousands of pairs of bacteria from two different collections of metabolic models, we find that the majority of pairs can exhibit competitive and cooperative interactions, depending on the resources available in the environment (Fig 2A). Despite this variability, we consistently find that cooperation is more common in environments with lower resource diversity. Thus even when pairs initially compete, as resources are consumed and removed pairs switch towards obligate forms of cooperation. These results highlight the dynamic nature of ecological interactions in response to environmental change.

A key result from our analyses is the high degree of plasticity in bacteria interactions depending on the environmental context. Previous studies have found differences in the relative frequencies of cooperative and competitive interactions across different environments [10,12,14,52–54]. Since the communities in these studies differ across environmental contexts it is difficult to disentangle whether it is the environment or specific pairs of species that are driving the changes in interactions. One recent study [19] focusing on a specific pair of bacteria, *Agrobacterium tumefaciens* and *Comamonas testosteroni*, has shown that such plasticity is possible as the pair shift between competition and cooperation depending on the concentration of linoleic acid. Through our large-scale metabolic approach we extend this finding and observe that plastic interactions are a common feature of bacteria pairs and should be the null expectation. This plasticity likely stems from the inherent complexity of metabolism in which the average metabolic network can import/export hundreds of different compounds from the environment. This complexity creates diverse opportunities for both competition and cooperation depending on the particular set of compounds available in the environment.

In spite of this complexity, we found some common trends in which environments select for certain types of interactions. Based on the competitive-exclusion principle, one might expect that resource-poor environments will favor competition over the limited availability of compounds [1]. At the opposite extreme, resource-rich environments may have a more diverse selection of compounds available for cross-feeding interactions to arise [37]. In contrast to our initial expectations, we found resource-poor environments were more likely to have cooperative interactions than competition, and we found that cooperation decreased in resource-rich environments (Fig 2C). Broadly our results do align with the stress gradient hypothesis which states that microbes tend to cooperate with each other under more 'stressful' conditions but compete more when the stress is reduced [27,55]. While the stress could manifest in different forms, e.g., toxicity [56] or low resource availability [56,57], in our system they manifest as a low diversity of resources, similar to [51].

Further supporting the stress gradient hypothesis, when compounds were removed from environments it ultimately led to obligate interactions regardless of whether the initial interactions were competitive or cooperative (Fig 6). In metabolic models from the CarveMe collection, $Fe^{2+}$ and $Fe^{3+}$ ions were major triggers, while in models of gut bacteria from the AGORA collection, many switches to obligacy were due to B vitamin removal. The case of B vitamins is intriguing as in the gut only approximately 12.5% of bacteria are predicted to produce all eight B vitamins [58] and the rest acquire them from the environment or via

complementary mechanisms with other bacteria [59,60]. Under conditions in which external B vitamins become rare or absent, the survival of these species will crucially depend on their mutually obligate partners or complete B vitamin producers. Thus, our findings linking resource-poor environments with obligate interactions may make sense as those environments may lack compounds that are essential for some bacteria but can be produced via cross-feeding [36,58,61,62]. Interestingly, in our random screens of environments, obligate interactions were the most common interaction type found—even in resource-rich environments—with over 99% of bacteria pairs having an obligate interaction in at least one environment. A previous metabolic modeling study also found frequent obligate interactions using a tailored algorithm on a set of 7 species [48]. Our analyses generalize this result and demonstrate that even an unbiased random search of environments can readily identify environments where mutual growth is only possible through obligate interactions. The relative ubiquity of obligate interactions has implications for the evolution of biological complexity, e.g., multicellularity or sociality, where obligacy is considered a promising initial state [2,63,64].

Across our analyses, we found that the difference between competitive and cooperative environments often came down to a single compound. Notably, a significant proportion of all transitions between cooperation and competition occurred through the removal of amino acids, emphasizing their role as modulators of microbial interactions [65–67]. The removal of terminal electron acceptors (e.g., oxygen, nitrate, and nitrite) also played a major role, as it forces bacteria to use alternative metabolic pathways with, e.g., anaerobic respiration and fermentation products [68]. We found the availability of other key cross-feeding intermediate metabolites like lactate, succinate, formate, or short chain fatty acids were less frequent causes of interaction switches, suggesting these might be less universal. Carbon sources, whose role on community composition is often investigated [37,51], are surprisingly rare among the top compounds initiating interaction switches (S2 and S3 Tables; with the exception of benzoate, which has also been highlighted by, e.g., [69]). We note that while certain classes of compounds were more often associated with switches between competition and cooperation, there was no single compound that could reliably distinguish between environments. This highlights that the composition of growth media as a whole plays a key role in shaping interactions.

Throughout this paper, we have used metabolic models to assess the effects of environmental resources on the ability of pairs of bacteria to compete or cooperate. While metabolic models have been used in a variety of other contexts [24,70–75], their development is still an active area of research [42,76–79]. In particular, we note that the models used in this paper do not feature any type of gene regulation. So although we can identify the potential for cooperation or competition, whether these interactions are realized or some other interaction (e.g., exploitation) actually happens will depend on a complex array of factors. One way we have tried to mitigate some of the variability from using metabolic approaches was to perform our analyses using two different model collections: AGORA which contains bacteria from the human gut and CarveMe which contains a more diverse collection of bacteria representing metabolisms across all (as sequenced in NCBI RefSeq release 84, [50]) bacterial life. Our main findings concerning the plasticity of bacterial interactions were consistent across the two collections. One instance where the results did systematically differ between collections is in the higher proportion of cooperative interactions observed in AGORA pairings compared with CarveMe. The difference in levels of competition between AGORA and CarveMe pairings is congruent with a recent study that found more cooperative potential between host-associated communities (such as the gut) than in free-living communities (such as in soil) [16,80]. As metabolic network analyses continue to improve, we expect that our ability to draw insights

and make predictions for specific pairings will be refined, though we expect our broad conclusions, that were consistent across both collections, are unlikely to change.

We also note that our use of flux balance analysis to infer metabolic interactions entails certain assumptions. For one, it focuses on those competitive and cooperative interactions that alter growth rate, specifically in actively growing populations. It also assumes that species (or metabolisms) are static and not evolving in ways that would change their metabolism. While metabolic models could be incorporated into an eco-evolutionary, dynamic framework [81], we do not do so here. Of course, cooperative/competitive interactions could occur in other contexts, such as survival under stress [82] or spatial constraints [83–85], and these interactions may themselves evolve over time [86,87]. We chose to use flux balance analysis and metabolic models to explore a well-defined context, where the only driver of selection is growth. Our results reveal that in the majority of cases the same pair of bacteria can exhibit either cooperation or competition, depending on relatively small changes in the resources present in an environment. Importantly, our metabolic modeling only demonstrates what interactions are possible— what actually occurs will depend on which specific reactions are used. Since a common feature in metabolic models is that there are many possible sets of reactions that can give rise to the same growth rate, this plasticity and how it is controlled by regulation will shape the outcome of interactions. This highlights the inherent plasticity of microbial interactions and raises the question of how eco-evolutionary forces determine the interactions that are ultimately realized.

Furthermore, here we report exclusively on pairwise interactions, but natural microbial communities often contain thousands of interacting species. This could result in higher-order interactions that lead to greater complexity and unpredictability of the interaction dynamics [88]. Similar to pairwise dynamics, though, higher-order interactions can result in increased cooperation with new species increasing beneficial interactions within the community [56,89]. For example, the addition of a third species has been shown to eliminate the antagonism between a pair of bacteria in a synthetic community of three Bacilota species [90]. Alternatively, additional species can result in adverse effects and increased competition [91], such as a switch from a cooperative to an antagonistic two-strain interaction in the presence of a third strain [92]. However, some studies show that pairwise co-cultures can partially predict interactions in more complex communities [12,93]. Therefore, assessing the types of pairwise species interactions across thousands of environments may provide insights into predicting overall community function.

Our results suggest a highly dynamic landscape of microbial interactions influenced by the availability of compounds in the environment. This finding has significant implications for our understanding of the eco-evolutionary dynamics of microbial communities. Traditionally, modeling approaches have relied on fixed interactions between pairs of species, such as those described by competitive Lotka-Volterra equations with constant interaction terms. However, our results indicate that a more complex framework, one that accounts for resource fluctuations and their effects on species interactions, may better predict community dynamics and even reveal new types of behaviors [8,94]. The highly plastic nature of these interactions also raises questions about how bacteria navigate them. Specifically, what strategies or heuristics have evolved to help bacteria capitalize on cooperative opportunities? Addressing these questions could illuminate how bacteria regulate their metabolisms to influence their ecological interactions. While our investigations into pairwise interactions provide significant insights into collective behavior, understanding the full impact of the environment on a community requires examining a broader network of interactions. Exploring the constraints and cooperative potential as microbial communities grow presents a fascinating avenue for further investigation.

## Methods

### Assessing growth of individuals and interactions of pairs

We obtained metabolic models of diverse prokaryotes in .xml format made available in ref. [95] for AGORA [17] and CarveMe [49] collections. We followed the procedure outlined in [43] to curate the metabolic models in a format for MATLAB R2022b. These models include a stoichiometric matrix with each column representing a reaction of the metabolism and rows representing chemical compounds (note, the models have been curated such that each row always corresponds to the same compound with names stored in a separate vector). Compounds are separated into two compartments: the extracellular compartment, and intracellular compartment (in AGORA this is simply the cytoplasm, while CarveMe also includes the periplasm separately). Multiplying the stoichiometric matrix by a vector of fluxes for the rate of flow through each reaction determines a right-hand-side vector of net changes in compound concentrations in each compartment. Models additionally include upper and lower bounds for all the reaction fluxes and the right-hand-side. It is assumed that there can be no accumulation of products in internal compartments and so the upper and lower bounds on the right-hand-side for chemical concentrations in internal compartments are 0, while non-zero bounds on extracellular compounds indicates that their concentration can be altered by the microbes present (note, there are no source or sink reactions in the stoichiometric matrix). Specifically, negative values of the right-hand-side for compounds in the extracellular compartment correspond to a loss from this compartment indicating they are imported by a microbe; positive values correspond to a gain of compounds in the extracellular compartment indicating that they have been exported by a microbe. Hence, we refer to compounds with negative lower bounds in the extracellular compartment as the 'environment'.

We use the metabolic models to determine the growth rate of microbes in different environments. To do this, we perform flux balance analysis and solve the associated linear program. For example, the linear program for the growth of an individual microbe $M_i$ in isolation is given by

$$
\begin{aligned}
\text{maximize} \quad & \lambda_{M_i} \\
\text{subject to} \quad & S_{M_i} \cdot \mathbf{v} = \mathbf{c} \\
& \mathbf{l_v} \leq \mathbf{v} \leq \mathbf{u_v} \\
& \mathbf{l_c} \leq \mathbf{c} \leq \mathbf{u_c},
\end{aligned}
\tag{1}
$$

where the objective function, $\lambda_{M_i}$, is the flux through the biomass reaction of microbe $M_i$ which is included in the models obtained. In this linear program $S_{M_i}$ is the stoichiometric matrix of microbe $M_i$; $\mathbf{v}$ is the vector of reaction fluxes which has lower bounds $\mathbf{l_v}$ and upper bounds $\mathbf{u_v}$; the right-hand-side vector $\mathbf{c}$ is the net change in compound concentrations bounded by $\mathbf{l_c}$ and $\mathbf{u_c}$. We solved this and subsequent linear programs by implementing the Gurobi optimization software [96] in MATLAB 2022b.

We determine the ecological interaction of a pair of bacteria in a particular environment by first computing the growth rate of each in isolation. These isolated growth rates serve as reference points against which we compare the growth rates when the bacteria grow together in a shared environment, where they can interact through the import/export of metabolites. The first environment we consider is based on the 'default' environment for each organism. In both collections, metabolic models come with an inbuilt default environment—lower and upper bounds on the right-hand-side vector ($\mathbf{l_c}$ and $\mathbf{u_c}$)—designed so that the corresponding microbe can grow in its default environment. By combining the default environments of two microbes we create a new environment that guarantees the growth of the pair and each

microbe in isolation. We create this combined environment (as in Fig 1) by summing the inbuilt values for $\mathbf{l_c}$ and $\mathbf{u_c}$ for the pair. Implementing these constraints in linear program (1) determines the growth rates of the individuals, $\lambda_{M_i}$ and $\lambda_{M_j}$.

For a pair, $M_i$ and $M_j$, we construct a combined stoichiometric matrix

$$S_{M_i+M_j} = \begin{pmatrix} S_{E_i} & S_{E_j} \\ S_{I_i} & 0 \\ 0 & S_{I_j} \end{pmatrix},$$

where we have rearranged the rows of $S_{M_i}$ for convenience such that $S_{M_i} = \left(S_{E_i}; S_{I_i}\right)$ with rows of $S_{E_i}$ corresponding to compounds in the extracellular compartment and rows of $S_{I_i}$ corresponding to the intracellular compartment. We define interactions between organisms by determining their maximal growth rate without harming (reducing the growth rate) of their partner when compared to their individual growth. For example, for pair $M_i$ and $M_j$, we compare the individual growth rate of $M_i$, $\lambda_{M_i}$, with its growth rate in the shared environment $\lambda_{M_i}^*$, determined by solving the linear program

$$\begin{aligned} \text{maximize} \quad & \lambda_{M_i}^* \\ \text{subject to} \quad & S_{M_i+M_j} \cdot \mathbf{v} = \mathbf{c} \\ & \mathbf{l_v} \leq \mathbf{v} \leq \mathbf{u_v} \\ & \mathbf{l_c} \leq \mathbf{c} \leq \mathbf{u_c}, \\ & \lambda_{M_j}^\dagger \geq \lambda_{M_j} - \epsilon, \end{aligned} \qquad (2)$$

where, $\mathbf{v} = \left(\mathbf{v}_{M_i}; \mathbf{v}_{M_j}\right)$ is the combined vector of reaction fluxes each bound the same as for the individuals; $\mathbf{c} = \left(\mathbf{c}_{E_i} + \mathbf{c}_{E_j}; \mathbf{c}_{I_i}; \mathbf{c}_{I_j}\right)$ is the right-hand-side vector with a shared extracellular compartment and separate intracellular compartments for $M_i$ and $M_j$. The linear program (2) for a pair includes an additional constraint to ensure that while optimizing the growth rate of individual $M_i$, the growth rate of the second species $(\lambda_{M_j}^\dagger)$ is no worse than when it was alone $(\lambda_{M_j}$, determined by solving (1) for $M_j$). Note, the inequality on the growth rate $\lambda_{M_j}^\dagger$ is calculated with a tolerance of $\epsilon = 0.001$. We solve the equivalent linear program for $M_j$ to determine its best possible growth rate, $\lambda_{M_j}^*$, without harming the growth of $M_i$. Table 1 indicates the different possible interactions between microbes $M_i$ and $M_j$. The exploitative interactions (+/-) or (-/+) have been left blank as such interactions are impossible under our optimization scheme: if we have a solution $\lambda_{M_i}^* > \lambda_{M_i}$ then by construction it must be the case that a $\lambda_{M_j}^\dagger \geq \lambda_{M_j}$ exists which would be a solution for $\lambda_{M_j}^*$. Indeed, in the metabolic modeling framework there is the potential for exploitation in both what we classify as competitive or cooperative environments but this depends on how resources are allocated and which specific reactions are used.

**Table 1. Table of potential interaction types for a pair of microbes $M_i$ and $M_j$. Equality of growth rates is calculated with a tolerance of $\epsilon = 0.001$, for example, $\lambda_{M_i} - \epsilon \leq \lambda_{M_i}^* \leq \lambda_{M_i} + \epsilon$.**

| | $\lambda_{M_j}^* < \lambda_{M_j}$ | $\lambda_{M_j}^* = \lambda_{M_j}$ | $\lambda_{M_j}^* > \lambda_{M_j}$ |
|---|---|---|---|
| $\lambda_{M_i}^* < \lambda_{M_i}$ | (-/-) | (-/=) | |
| $\lambda_{M_i}^* = \lambda_{M_i}$ | (=/-) | (=/=) | (=/+) |
| $\lambda_{M_i}^* > \lambda_{M_i}$ | | (+/=) | (+/+) |

## Essential compounds for pairs

When searching different environments we wanted to minimize the number of environments sampled where the microbes cannot grow together, without otherwise biasing the sampling of environmental compounds. To do so, first we identified a set of environmental compounds that were 'essential' to both of the microbes with no possible substitutions. We did this by growing the microbes together in a replete environment where every possible compound in the extracellular compartment was provided (had negative lower bounds). We then individually removed each compound from the replete environment (setting its lower bound to 0) and tested whether both microbes were able to grow together ($\lambda^*_{M_i} > \epsilon$ and $\lambda^*_{M_j} > \epsilon$, where $\epsilon = 0.1$ is a small growth tolerance). If the pair were not able to grow together when a particular compound was removed, this compound was identified as 'essential' as no possible substitutions exist to replace it. When constructing environments, we always provided the essential compounds for that pair (see S1 Fig). However, we note that the set of essential compounds was not necessarily sufficient—other compounds might be necessary for the growth of the microbe pair but potential substitutions exist which need to be provided. See S2 Fig for a histogram of the number of essential compounds per pair and S1 Table for the 10 most common essential compounds across the 10,000 pairs we considered.

## Algorithm for generating environments

We generated different environments by changing the bounds of the right-hand-side vector $\mathbf{c}$ ($\mathbf{l_c}$ and $\mathbf{u_c}$). As positive values of $\mathbf{c}$ indicate the export of compounds into the environment, we kept the upper bounds on the right-hand-side as the sum of default values for the pair throughout all our analyses. Changing the lower bounds on $\mathbf{c}$ we were able to explore the effects of the availability of different environmental compounds on microbial interactions.

In this analysis we focused on the presence and absence of compounds in the environment and their effect on the interaction between pairs of microbes. As such, when generating environments, we set $\mathbf{l_c}$ = –1000mmol gDW$^{-1}$ h$^{-1}$ for compounds present in the environment; this is the default value for a single microbe and so deemed to be a value sufficient for growth but with the potential to induce competition between the pairs. By contrast, if both microbes included the same compound in their default environments, we would set $\mathbf{l_c}$ = –2000mmol gDW$^{-1}$ h$^{-1}$ when considering the interaction in their combined default environment. Compounds not present in the environment had $\mathbf{l_c}$ = 0. We additionally repeated the analysis for 10,000 pairs with a reduced concentration of –500mmol gDW$^{-1}$ h$^{-1}$ and obtained similar results (see S21 Fig). Of course, further lowering the concentration of compounds, we expect to generate higher degrees of competition between pairs.

Each pair was always provided with its essential compounds (identified by *Methods: Essential compounds for pairs*). In addition, we randomly sampled a fixed number of additional compounds (50 or 100) uniformly from those extracellular compounds that could be used by at least one of the pair. These usable compounds were identified as those which appeared in at least one reaction in the combined stoichiometric matrix. On average, there were 177 usable compounds per pair in AGORA and 206 usable compounds per pair in CarveMe (see S22 Fig), hence we sampled fixed size environments of 50 or 100 additional compounds. When selecting 10,000 random pairings in each collection, we had to discard 317 randomly selected pairs from AGORA and 58 from CarveMe as their combined environment included less than 100 compounds once the essential compounds had been removed.

### Single removal of compounds

To determine the stability of interactions to the loss of a single compound from a given environment for a particular pair of bacteria, we computed the interaction when each of the additional compounds was individually removed from the environment. As essential compounds are essential to that pair in any environment (with no possible substitutes), we did not remove these compounds. We went through each of the additional compounds one-by-one removing them from the complete starting environment (essential compounds plus all 100 additional compounds) and determining the effect of removing it from the environment on the interaction between the pair. This process was repeated for each compound.

### Degradation of environments

To determine the sensitivity of interactions to perturbations of the environment, we first examined whether the interaction between 1,000 different pairs would change upon the removal of any single compound from either competitive or facultative cooperative environments. We then wanted to explore the plasticity of interactions between pairs by removing the additional (substitutable) compounds from both competitive and cooperative starting environments until at least one organism was no longer able to grow. As only some of the available compounds are imported by the pair (i.e. only some of the environmental compounds have negative values for $c_E$ when the linear problem (2) is solved), we removed any redundant compounds from the initial environment that were not imported by either of the microbes (see S1 Fig for a schematic illustrating the distinction between 'used' and 'redundant' compounds). From this reduced environment, we then sequentially removed compounds until at least one organism was no longer able to grow. We did not remove essential compounds as it was already known that this would terminate the sequence. (Note, that while the other compounds are not essential-without-substitutes, once the environment is reduced such that no substitute remains, at least one of the microbes can no longer grow.) For each pair considered, we tried two different starting environments, one competitive and one facultative cooperative, and 50 different random orderings of compound removal.

## Supporting information

**S1 Fig. Construction of environments.** Schematic shows the algorithm for construction of environments for pairs of microbes. First compounds are identified that appear in the stoichiometric matrix of at least one microbe in that pair. These 'usable' compounds are then individually removed from the replete environment with all other compounds present; if the pair cannot grow together without this compound then it is labeled as 'essential', other compounds are substitutable. Environments always contain the essential compounds identified for that pair, additionally $N$ substitutable compounds are randomly selected ($N = 50$ for smaller environments and $N = 100$ for larger environments). If the pair grow in a particular environment, flux balance analysis outputs fluxes of environmental compounds. Compounds with negative fluxes have a net uptake from the environment and are 'used' by the microbe(s). While these flux outputs are not unique, the un-used compounds could be considered 'redundant'.
(EPS)

**S2 Fig. Essential compounds per pair.** Histograms show the number of essential compounds for 10,000 pairs in AGORA (A) and CarveMe (B) collections, as used in the main paper.
(EPS)

**S3 Fig. Correlation of competition or cooperation with metabolic complexity of bacteria pairs; measured by number of metabolites.** A and B) Histograms show the frequency of number of metabolites that could be used per bacteria in the AGORA (A) and CarveMe (B) collections, which can be used as a proxy for metabolic complexity. Dashed lines split the data into thirds which are used to classify 'low', 'medium', and 'high' complexity in the following plots. C–F) Box plots of the number of competitive (C and D) and cooperative (E and F) interactions per pair analyzed in AGORA (C and E) and CarveMe (D and F). In all cases the classification of data by metabolic complexity with number of metabolites is statistically significantly different than if the classification was done randomly from a uniform distribution ($p$-$value$ <0.001). Stars are used to indicate whether the complexity classification with the highest and lowest observations of competitive/cooperative environments were significantly different from all other classifications using an ANOVA multiple comparison test (*** $p <$ 0.001). When both species have a low metabolic complexity, we found the highest number of competitive and fewest cooperative environments. The fewest competitive environments were found between low–high pairs, and the most cooperative environments were found for pairs with at least one high complexity species.
(EPS)

**S4 Fig. Correlation of competition or cooperation with metabolic complexity of bacteria pairs; measured by number of reactions.** A and B) Histograms show the frequency of number of reactions per bacteria in the AGORA (A) and CarveMe (B) collections, which can be used as a proxy for metabolic complexity. Dashed lines split the data into thirds which are used to classify 'low', 'medium', and 'high' complexity in the following plots. C–F) Box plots of the number of competitive (C and D) and cooperative (E and F) interactions per pair analyzed in AGORA (C and E) and CarveMe (D and F). In all cases the classification of data by metabolic complexity with number of reactions is statistically significantly different than if the classification was done randomly from a uniform distribution ($p$-$value$ <0.001). Stars are used to indicate whether the complexity classification with the highest and lowest observations of competitive/cooperative environments were significantly different from all other classifications using an ANOVA multiple comparison test (*** $p <$ 0.001 and ** $p <$ 0.01). The trends for cooperative environments with metabolic complexity here qualitatively agree with Fig S3E and S3F where complexity was measured by number of metabolites. However, the highest number of competitive environments was now found between high–high pairs and the pairings with fewest competitive environments differed between the two collections.
(EPS)

**S5 Fig. Compound appearance in competitive *vs* cooperative environments.** A and B) Plots show the number of appearances of each compound in competitive vs cooperative environments in AGORA (A) and CarveMe (B). Grey dashed line indicates the proportion of competitive to cooperative environments found (11.2% in AGORA and 25.6% in CarveMe); red dots show the real data from the environments found (939656 environments in AGORA and 804379 in CarveMe); blue dots show simulation data if the identified environments are randomly assigned competitive or cooperative with a probability given by the proportion of such environments found overall. The majority of compounds (97.4% AGORA, 95.8% CarveMe) appeared in a higher proportion of competitive or cooperative environments than by independent random sampling ($p$-$value$ <0.001). C and D) Plots show the cumulative distribution function of compound appearance biases in AGORA (C) and CarveMe (D); 68.2% compounds have a less that 20% bias away from equal probability of competitive or cooperative

environments in AGORA and 78.4% in CarveMe. These show that the median bias is only 12.1% in AGORA and 8.0% in CarveMe.
(EPS)

**S6 Fig. Amino acid appearance in competitive *vs* cooperative environments.** A and B) Histograms show the number of amino acids per competitive, facultative cooperative and obligate environment in AGORA (A) and CarveMe (B). (An ANOVA confirms there is a statistically significant difference between the number of amino acids in competitive vs cooperative environments, $p$-value <0.001.) In AGORA there is on average 1 more amino acid in competitive environment than in either obligate or facultative cooperative environments (14 vs 13), while in CarveMe there are an average of 9 amino acids in all the types of environments. C and D) Bar charts show the percentage of competitive, facultative cooperative and obligate environments that each amino acid appears in in AGORA (C) and CarveMe (D). In AGORA, each amino acid consistently appears in a higher proportion of competitive environments than facultative cooperative or obligate, whereas CarveMe does not have this consistent trend. There is also a higher variation in the proportion of environments that each amino acid appears in CarveMe.
(EPS)

**S7 Fig. Simple sugar appearance in competitive *vs* cooperative environments.** A and B) Histograms show the number of simple sugars—aldoses and ketoses—per competitive, facultative cooperative and obligate environment in AGORA (A) and CarveMe (B). An ANOVA confirms there is a statistically significant difference between the number of simple sugars in competitive vs cooperative environments, $p$-value <0.001. However, to the nearest integer, in both collections there is an average of 4 aldoses/ketoses in all types of environments. C and D) Bar charts show the percentage of competitive, facultative cooperative and obligate environments that each aldose/ketose appears in in AGORA (C) and CarveMe (D). Each individual sugar also appears in similar proportions of each type of environment.
(EPS)

**S8 Fig. Similarity across competitive and cooperative environments.** A-D) Bar charts show the mean Jaccard distance between competitive, cooperative and competitive vs cooperative environments from AGORA (A) smaller and B) larger environments) and CarveMe (C) smaller and D) larger environments). The means have been taken from 20 bins of 1,000 random samples of each type of environment. Identical environments would have a Jaccard distance of 0, whereas completely distinct environments would have a Jaccard distance of 1; we find the mean Jaccard distances of 0.5–0.82 between environments indicating we found a diverse spread of environments where bacteria can compete or cooperate with moderate overlap. While there is a statistically significant difference (ANOVA $p$-value <0.001) in the distance between environments of the same type (purely competitive or cooperative) and comparing across both types of environment (competitive vs cooperative), this difference is small (< 4%) indicating that it would be difficult to distinguish between competitive and cooperative environments with a simple similarity metric. E-H) Box plots show the mean Jaccard distance between competitive, cooperative and competitive vs cooperative environments individual pairs. Data is shown for all pairs where we found at least five examples of both competitive and cooperative environments in AGORA (E) smaller and F) larger environments) and CarveMe (G) smaller and H) larger environments). On average, the Jaccard distance between environments for a particular pair is smaller than the distance between all the competitive and cooperative environments we found indicating that the environments for a particular pair are more similar than the global average. However, an ANOVA shows that there is no significant

difference (all $p$-values > 5%) in the Jaccard distance between the same type of environment and competitive vs cooperative. This suggests that it is not possible to distinguish between competitive and cooperative environments with a simple similarity metric on a pair-by-pair basis.
(EPS)

**S9 Fig. Compounds that cause switches in interactions between 1000 pairs in AGORA.** Blue bars show the data for compounds from environments generated by our search algorithm. Red dots show the expected number of switches that would be caused if it was by random chance; variation arises from the different number of environments that compounds are present in. A) Switches from competitive to facultative cooperative environments are caused by 38% of compounds. B) Switches from facultative cooperative to competitive environments are caused by 30% of compounds. C) Switches from competitive to obligate environments are caused by 23% of compounds. D) Switches from facultative cooperative to obligate environments are caused by 25% of compounds. Across these figures, only 45% compounds are responsible for all the switches observed between competitive and cooperative environments.
(EPS)

**S10 Fig. Compounds that cause switches in interactions between 1000 pairs in CarveMe.** Blue bars show the data for compounds from environments generated by our search algorithm. Red dots show the expected number of switches that would be caused if it was by random chance; variation arises from the different number of environments that compounds are present in. A) Switches from competitive to facultative cooperative environments are caused by 38% of compounds. B) Switches from facultative cooperative to competitive environments are caused by 39% of compounds. C) Switches from competitive to obligate environments are caused by 28% of compounds. D) Switches from facultative cooperative to obligate environments are caused by 34% of compounds. Across these figures, 58% compounds are responsible for all the switches observed between competitive and cooperative environments.
(EPS)

**S11 Fig. Amino acid-driven switches between interactions.** A and B) Bar charts show the percentage of environments where each amino acid was present and its removal caused a switch from competition (A) or facultative cooperation (B) to facultative cooperation, competition, or obligate interactions for the 1,000 pairs in AGORA where a single-compound removal sensitivity analysis performed in Fig 4A and 4B. C and D) Bar charts show the same analysis of amino acid-driven switches between interactions as in A and B but for data from the 1,000 pairs in CarveMe in Fig 4C and 4D. The stars above bars indicate cases where the removal of the amino acid resulted in statistically significantly ($p$-value <0.01) more switches than if it was equally likely that any compound in the environment could cause the switches observed in Fig 4. The majority of amino acids are responsible for a significant number of switches to obligate interactions when removed, approximately half are also responsible for a significant number of switches between competitive and facultative cooperative interactions.
(EPS)

**S12 Fig. Simple sugar-driven switches between interactions.** A and B) Bar charts show the percentage of environments where simple sugars (aldoses and ketoses) were present and their removal caused a switch from competition (A) or facultative cooperation (B) to facultative cooperation, competition, or obligate interactions for the 1,000 pairs in AGORA where a single-compound removal sensitivity analysis performed in Fig 4A and 4B. C and D) Bar charts show the same analysis of simple sugar-driven switches between interactions as in A

and B but for data from the 1,000 pairs in CarveMe in Fig 4C and 4D. The stars above bars indicate cases where the removal of the sugar resulted in statistically significantly (*p*-value <0.01) more switches than if it was equally likely that any compound in the environment could cause the switches observed in Fig 4. Removing simple sugars from the environment rarely causes a switch in the interaction, especially in comparison to amino acids (shown in Fig S11). However, a few sugars are responsible for a significant number of switches both from competitive to facultative cooperative and from facultative cooperative to competitive. (EPS)

**S13 Fig. Interactions between common aquatic bacteria as compounds are removed from the environment.** Columns in the heatmaps show the interactions between pairs of bacteria as compounds are removed; going down the rows subsequent compounds are removed from the environment. Each column is a different ordering of compound removal; columns have been sorted by length of time to the first change in interaction. Colors correspond to different interactions: competition in pink; facultative cooperation in light green; one-way obligate interactions in green; two-way obligate interactions in dark green; no growth in black and other interactions in gray ( (+/ =) or (x/ =) ). In the left panels pairs begin in an environment for competition and in right panels for facultative cooperation. A) and B) *Prochlorococcus_marinus_str_MIT_9312* and *Pseudomonas_aeruginosa_PAO1*, C) and D) *Prochlorococcus_marinus_str_MIT_9312* and *Alteromonas_macleodii_ATCC_27126*.
(EPS)

**S14 Fig. Interactions between common soil bacteria as compounds are removed from the environment.** In the left panels pairs begin in an environment for competition and in right panels for facultative cooperation. A) and B) *Bradyrhizobium_japonicum_USDA_6* and *Sphingomonas_melonis_TY*, C) and D) *Bradyrhizobium_japonicum_USDA_6* and *Methylocystis_parvus_OBBP*, E) and F) *Bradyrhizobium_japonicum_USDA_6* and *Actinoplanes_friuliensis_DSM_7358*, G) and H) *Bradyrhizobium_japonicum_USDA_6* and *Blastococcus_endophyticus_DSM_45413*, I) and J) *Bradyrhizobium_japonicum_USDA_6* and *Azospirillum_brasilense_Sp_7*, K) and L) *Bradyrhizobium_japonicum_USDA_6* and *Rubrobacter_xylanophilus_DSM_9941*, M) and N) *Bradyrhizobium_japonicum_USDA_6* and *Mycobacterium_rhodesiae_JS60*, O) and P) *Bradyrhizobium_japonicum_USDA_6* and *Physisphaera_mikurensis_NBRC_102666*, Q) and R) *Bradyrhizobium_japonicum_USDA_6* and *Streptomyces_coelicolor_A3_2*.
(EPS)

**S15 Fig. Interactions between common gut bacteria as compounds are removed from the environment (CarveMe).** In the left panels pairs begin in an environment for competition and in right panels for facultative cooperation. A) and B) *Prevotella_copri_DSM_18205* and *Bacteroides_vulgatus_ATCC_8482*, C) and D) *Prevotella_copri_DSM_18205* and *Lactobacillus_fermentum_IFO_3956*, E) and F) *Prevotella_copri_DSM_18205* and *Collinsella_aerofaciens_ATCC_25986*. We note that in some cases the set of compounds essential to the starting environment is sufficient for the growth of the pair, as seen in (E).
(EPS)

**S16 Fig. Interactions between common gut bacteria as compounds are removed from the environment (AGORA).** In the left panels pairs begin in an environment for competition and in right panels for facultative cooperation. A) and B) *Prevotella_copri_DSM_18205* and *Bacteroides_vulgatus_ATCC_8482*, C) and D) *Prevotella_copri_DSM_18205* and *Clostridium_perfringens_ATCC_13124*, E) and F) *Prevotella_copri_DSM_18205* and *Ruminococcus_gnavus_AGR2154*, G) and H) *Prevotella_copri_DSM_18205* and

*Eubacterium_rectale_ATCC_33656*, I) and J) *Prevotella_copri_DSM_18205* and *Bifidobacterium_longum_longum_JCM_1217*, K) and L) *Prevotella_copri_DSM_18205* and *Escherichia_coli_O157_H7_str_Sakai_Sakai_substr_RIMD_0509952*.
(EPS)

**S17 Fig. Interactions between gut bacteria *Akkermansia* and *Blautia* species as compounds are removed from the environment (CarveMe).** In the left panels pairs begin in an environment for competition and in right panels for facultative cooperation. A) and B) *Akkermansia_muciniphila_ATCC_BAA_835* and *Blautia_hansenii_DSM_20583*, C) and D) *Akkermansia_muciniphila_ATCC_BAA_835* and *Blautia_hydrogenotrophica_DSM_10507*, E) and F) *Akkermansia_muciniphila_ATCC_BAA_835* and *Blautia_obeum_2789STDY5608838*, G) and H) *Akkermansia_muciniphila_ATCC_BAA_835* and *Blautia_obeum_ATCC_29174*, I) and J) *Akkermansia_muciniphila_ATCC_BAA_835* and *Blautia_producta_ATCC_27340_DSM_2950*, K) and L) *Akkermansia_muciniphila_ATCC_BAA_835* and *Blautia_schinkii_DSM_10518*.
(EPS)

**S18 Fig. Interactions between gut bacteria *Akkermansia* and *Blautia* species as compounds are removed from the environment (AGORA).** In the left panels pairs begin in an environment for competition and in right panels for facultative cooperation. A) and B) *Akkermansia_muciniphila_ATCC_BAA_835* and *Blautia_hansenii_VPI_C7_24_DSM_20583*, C) and D) *Akkermansia_muciniphila_ATCC_BAA_835* and *Blautia_hydrogenotrophica_DSM_10507*, E) and F) *Akkermansia_muciniphila_ATCC_BAA_835* and *Blautia_obeum_ATCC_29174*, G) and H) *Akkermansia_muciniphila_ATCC_BAA_835* and *Blautia_producta_DSM_2950*, I) and J) *Akkermansia_muciniphila_ATCC_BAA_835* and *Blautia_wexlerae_DSM_19850*.
(EPS)

**S19 Fig. Environmental degradation summaries with all paths included.** Diagrams show the full summary of interaction changes between 300 pairs of bacteria as compounds are removed from the environment as in Fig 6C–6F. The left column summarizes interaction paths starting in competition ((A) AGORA and (C) CarveMe), and the right starting in facultative cooperation (B) AGORA and (D) CarveMe). The semi-circular grid indicates the number of interaction switches. Nodes and line thickness are proportional to the square-rooted proportion of interactions taking that path; additionally paths are sorted such that the most common branch at each node is furthest to the left. All paths are included here; it was possible to observe 10 switches between interactions as the environment degraded.
(EPS)

**S20 Fig. Environmental degradation first switches *vs* fixed transition probabilities.** Bar charts summarize the first transitions from initial starting environments from Fig 6 in dark gray bars (A and B show data from AGORA and C and D from CarveMe; A and C start in competitive environments and B and C start in cooperative environments). By contrast, light gray bars show data for the same number of simulations but here the probability that a switch occurs is based on fixed probabilities calculated determined from the single removal of compounds (Fig 4); the new simulations were run until we observed an interaction switch or all 'substitutable' (not always essential) compounds had been removed from the environment. An ANOVA confirms that the new simulations are statistically significantly different from the data in Fig 6 (*p*-value <0.001). When environments degrade we find more cases of obligate interactions than with the fixed probabilities and fewer cases where neither bacteria can grow.
(EPS)

**S21 Fig. Environment-driven variation in competition and cooperation with available concentration flux -500mmol gDW$^{-1}$ h$^{-1}$.** Figure shows the same analysis as Fig 2 in the main paper but with a reduced value for the lower bound of the concentration flux (right-hand-side of the flux balance analysis) of environmentally available compounds; where we set the right-hand-side lower bound to -1000mmol gDW$^{-1}$ h$^{-1}$ in the main paper, we here use -500mmol gDW$^{-1}$ h$^{-1}$. A and B) Bar charts show the proportions of bacteria pairs where it was possible to find at least one environment for potential cooperation and/or competition in AGORA (A) and CarveMe (B). We now find at least one competitive and one cooperative environment for 59% (standard deviation 1.6%) pairs in AGORA and 90% pairs (standard deviation 0.9%) in CarveMe. C–F) Triangle plots show the proportion of different interactions across the 100 viable growth environments identified for pairs of bacteria from (A) and (B). Each point corresponds to a pair of bacteria, with the relative distance to each vertex on the triangle representing the proportion of environments for competitive, cooperative and other interactions identified for that pair. The darkest shading indicates the highest density of points on each plot. Left panel (C and E) shows results for AGORA and right panel (D and F) for CarveMe. Smaller environments (essential compounds plus 50 compounds) are shown in (C) and (D) and larger environments (essential compounds plus 100 compounds) are shown in (E) and (F). As in the main paper, we found a more diverse spread of interactions in larger environments.
(EPS)

**S22 Fig. Potential size of environments per pair.** A and B) Histograms show the number of environmental compounds in combined default environments for 10,000 pairs in AGORA (A) and CarveMe (B) collections, as used in the main paper Fig 1B. C and D) Histograms show the total number of environmental compounds that could be used by the same 10,000 pairs in AGORA (C) and CarveMe (D) collections. This is the number of environmental compounds that at least one of the bacteria contain within a metabolic reaction. Default environments contain almost all useable environmental compounds hence the histograms (A) and (B) look similar to (C) and (D).
(EPS)

**S1 Table. Common essential compounds.** The 10 most common essential compounds for pairs of bacteria in AGORA (a) and CarveMe (b).
(PDF)

**S2 Table. Compounds which cause transitions in AGORA.** Top ten compounds responsible for transitions a) competition to facultative cooperation; b) facultative cooperation to competition; c) competition to obligate; d) facultative cooperation to obligate in AGORA pairs considered in S9 Fig.
(PDF)

**S3 Table. Compounds which cause transitions in CarveMe.** Top ten compounds responsible for transitions a) competition to facultative cooperation; b) facultative cooperation to competition; c) competition to obligate; d) facultative cooperation to obligate in CarveMe pairs considered in S10 Fig.
(PDF)

**S1 Appendix. Mechanistic interpretation of transitions in case study Fig 5.**
(PDF)

## Acknowledgments

The authors thank the referees and editors for helping to improve the paper and IceLab for providing a supportive and collaborative environment for interdisciplinary science.

## Author contributions

**Conceptualization:** Josephine Solowiej-Wedderburn, Jennifer T. Pentz, Ludvig Lizana, Bjoern O. Schroeder, Peter A. Lind, Eric Libby.

**Data curation:** Josephine Solowiej-Wedderburn.

**Formal analysis:** Josephine Solowiej-Wedderburn, Eric Libby.

**Funding acquisition:** Ludvig Lizana, Bjoern O. Schroeder, Peter A. Lind, Eric Libby.

**Investigation:** Josephine Solowiej-Wedderburn, Eric Libby.

**Methodology:** Josephine Solowiej-Wedderburn, Eric Libby.

**Project administration:** Eric Libby.

**Supervision:** Eric Libby.

**Validation:** Josephine Solowiej-Wedderburn.

**Visualization:** Josephine Solowiej-Wedderburn, Jennifer T. Pentz, Eric Libby.

**Writing – original draft:** Josephine Solowiej-Wedderburn, Jennifer T. Pentz, Eric Libby.

**Writing – review & editing:** Josephine Solowiej-Wedderburn, Jennifer T. Pentz, Ludvig Lizana, Bjoern O. Schroeder, Peter A. Lind, Eric Libby.

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
