## [Decision Letter · Decision Letter 0]

26 Jun 2025

PCOMPBIOL-D-24-01869

Competition and cooperation: the plasticity of bacterial interactions across environments

PLOS Computational Biology

Dear Dr. Libby,

Thank you for submitting your manuscript to PLOS Computational Biology. After careful consideration, we feel that it has merit but does not fully meet PLOS Computational Biology's publication criteria as it currently stands. Therefore, we invite you to submit a revised version of the manuscript that addresses the points raised during the review process.

Please submit your revised manuscript within 60 days Mar 24 2025 11:59PM. If you will need more time than this to complete your revisions, please reply to this message or contact the journal office at ploscompbiol@plos.org. Please include the following items when submitting your revised manuscript:

We look forward to receiving your revised manuscript.

Kind regards,

Chaitanya S Gokhale, PhD

Academic Editor

PLOS Computational Biology

Natalia Komarova

Section Editor

PLOS Computational Biology

**Additional Editor Comments:**

Dear Eric,

I have been looking forward to getting replies from additional reviewers but to no avail. It has been really tricky to get reviewers and then their responses in time.

In the meantime I would like to send back the reports that we have and in view of the reports I would recommedn going through the draft very thoroughly.

In my opinion this is really good work and it has the chance to be very clear once the quesries of the reviewers are addressed. The reviewers have done a great job in thoroughly going through the manuscript and I support their views.

The manuscript lacks some clarifications and would benefit from addressing the issues. Some of the major issues are conceptual but others also relate to the methods used and their opacity.

E.g. both the reviewers in their major point #2 address teh issue of how to optimally address the environment similarity dissimilarity measure but lacks a metric. Making definitions clear and stating them upfront along with the assumptions also seems to be common point of issue for both the reviewers which I would tend to agree too with my reading of the manuscript. Lastly I would urge to make clear the statisical medhods and perhaps deposit them along side the other code and data in a repository that supports doi registration. As Reviewer 2 mentions, Zenodo might be an option.

Given a detailed reply to the reviewers comments I believe the manuscript will definitely benefit in an increase in clarity and rigor.

Again I apologise for the delay.

All the best.

**Journal Requirements:**

3) Please ensure that the funders and grant numbers match between the Financial Disclosure field and the Funding Information tab in your submission form. Note that the funders must be provided in the same order in both places as well.

**Reviewers' comments:**

Reviewer's Responses to Questions

**Comments to the Authors:**

Reviewer #1: Review of the Paper: "Competition and cooperation: The plasticity of bacterial interactions across environments"

Authors: Josephine Solowiej-Wedderburn, Jennifer T. Pentz, Ludvig Lizana, Bjoern O. Schroeder, Peter Lind, Eric Libby

Manuscript Number: PCOMPBIOL-D-24-01869

---

The manuscript provides an innovative computational framework to explore the plasticity of microbial interactions across diverse environmental conditions using genome-scale metabolic models. Analyzing 10,000 bacterial pairs across thousands of simulated environments from AGORA and CarveMe databases, the authors reveal that resource depletion often drives interactions toward obligate cooperation. These findings are valuable for understanding microbial ecology and synthetic community design. However, the manuscript would greatly benefit from addressing several key methodological and conceptual gaps, particularly by expanding the perturbation analysis and integrating it with metrics of environmental similarity. Below, I outline the major and minor points requiring attention to strengthen the manuscript.

---

Major Points

1. Perturbation Analysis Framework

The current perturbation analysis focuses on the removal of single compounds to assess the robustness of microbial interactions. However, this approach does not differentiate between redundant and non-redundant compounds. Removing a redundant compound may have minimal ecological impact, whereas removing an essential, non-redundant compound could drastically alter interaction dynamics. Incorporating redundancy metrics—based on functional or structural similarity—would provide a more realistic and biologically meaningful framework for perturbation analysis.

Furthermore, this analysis could naturally integrate with the concept of environmental similarity (point 2 below). By characterizing environments based on molecular similarity or dissimilarity, the study could explore whether interactions are more resilient to changes in chemically similar environments versus divergent ones. This connection would allow the authors to investigate whether resilience is a function of environmental redundancy or diversity. Expanding the perturbation analysis in this manner would add significant depth and broaden the study's implications.

2. Environmental Similarity and Diversity

While the manuscript emphasizes the impact of environmental diversity (i.e., the number of compounds) on microbial interactions, it neglects the molecular composition of these environments. Chemically similar environments may yield comparable interaction dynamics, while divergent environments might drive distinct patterns. Introducing a metric for environmental "distance" (e.g., based on molecular functional groups or structural similarity) would provide new insights into how microbial interactions respond to environmental variability.

Integrating environmental similarity metrics with the perturbation framework could enable the authors to test whether resilience and shifts toward cooperation are mediated by compound-level redundancies or larger-scale compositional features. Such an analysis would significantly enhance the biological relevance of the findings.

3. Definitions of Cooperation and Competition

The manuscript defines cooperation as mutual benefit, while competition only requires one organism to experience a negative effect. This asymmetry risks conflating competition with exploitative interactions, where one organism benefits at the expense of another. Exploitative interactions are distinct from pure competition and cooperation, and failing to separate these categories may skew the analysis and interpretation. Providing clear definitions and explicitly distinguishing exploitative interactions would improve the rigor of the study.

4. Beyond Pairwise Interactions

The study focuses exclusively on pairwise interactions, overlooking the complexity of multi-species microbial communities. Multi-species interactions often involve indirect effects and higher-order dynamics that could shift the balance between cooperation and competition. For instance, larger communities might favor cooperation through metabolic complementarity or increase competition due to overlapping resource needs. A discussion of these dynamics, supported by relevant literature, would add critical ecological context to the findings.

5. Statistical Rigor and Transparency

The manuscript provides limited detail on the statistical framework employed, including whether multiple comparisons were corrected or assumptions were validated. Given the scale of the data, statistical robustness is crucial. The authors should outline their methods in detail, including any corrections, sensitivity analyses, and justifications for sample sizes and thresholds. This transparency will enhance the credibility of the findings.

---

Minor Points

1. Terminology and Conceptual Clarity

The manuscript occasionally conflates terms such as competition and cooperation with metabolic cross-feeding and resource competition. Since cross-feeding does not always imply cooperation, clear and consistent definitions throughout the text are necessary to avoid confusion.

2. Visualizations

Some figures, particularly the triangular plots (e.g., Figures 2 and 3), are dense and challenging to interpret. Simplifying these visualizations, adding annotations to highlight key trends, and providing more detailed figure legends would improve accessibility.

3. B-Vitamin Dependencies

B-vitamin dependencies are mentioned but not fully explored in the discussion. Highlighting the ecological implications of these dependencies, particularly in shaping obligate cooperation in host-associated environments, would strengthen the manuscript.

4. Redundancy in Metabolic Models

The models do not appear to account for enzymatic or pathway redundancy, which could buffer microbial interactions against environmental perturbations. Including a brief discussion of how metabolic redundancy influences interaction dynamics would provide additional depth.

5. Reproducibility and Data Accessibility

While the manuscript references a GitHub repository for code, it is unclear whether raw data and intermediate analysis outputs are also available. Ensuring full reproducibility by providing these resources is essential.

6. Compound-Specific Insights

Certain compounds, such as oxygen and nitrate, are identified as key modulators of interaction shifts. Elaborating on the mechanistic basis for these effects would provide additional depth to the discussion.

7. Language and Style

The manuscript occasionally uses jargon-heavy language that could deter readers. Simplifying such phrases without losing scientific rigor would improve readability.

Reviewer #2: In their manuscript "Competition and cooperation: the plasticity of bacterial interactions across environments", Solowiej-Wedderburn J et al. use two metabolic network databases to investigate the effect of growing pairs of microbes together. Such databases only contain the metabolic information of individual microbes. Thus, Solowiej-Wedderburn J et al. theoretical approach relies on joining those individual metabolisms and measuring the effect on their maximum individual growth rate (hereafter interaction). By doing so, they are faced with the question of which extracellular compounds (hereafter environment) are essential to sustaining the pair's life. They test a large and random set of compounds, finding that growth rates can be affected in all sorts of ways, including cooperation (+/+), competition (-/-) and others, including neutral (=/=). Their high-throughput approach is sound and addresses an important question with implications for microbial ecology and evolution. I enjoyed reading this work, which would be a valuable contribution to PLoS Computational Biology. However, some model assumptions and implications seem to require further elaboration. Find a point-by-point list below.

Major.

1. Defining cooperation and competition is essential. However, that these are linked to the growth rate is clarified too late. I suggest defining them from the introduction and talking about the growth rate earlier (abstract and introduction), as readers expect to find this information early on. Actually, referring to growth rate seems more natural than referring to cooperation and competition (more ambiguous terms) sometimes. Also, you could mention the pros/cons of using growth rate to define cooperation/competition and whether your results are robust to this choice.

2. Your approach scans an impressive number of environments and microbial pairs. However, early on, one wonders about the assumptions made and their implications when comparing those different environments and microbial pairs. For example, you could mention that temperature, pH, and gene/expression regulation are not considered.

3. It would be helpful to make it more clear whether your model is dynamic or if it assumes an equilibrium. This is particularly important for the section where you degrade the environment one compound at a time. Currently, many of the model assumptions and critical details are restricted to the methods section, but they are important for interpreting results and discussion at times.

4. Like other metabolic models, you assume the growth rate is maximized. This assumption should be stated more visibly. In fact, you could mention more model limitations, assumptions or biases in your discussion.

5. The magnitude of your work offers an excellent opportunity for an ample discussion about the eco-evolutionary implications of cooperation and competition. This would be very satisfying to see for many readers.

Minor.

6. In your abstract, please make it clear that this is a theoretical work.

7. You could briefly mention alternative explanations for the plasticity of interactions apart from the environment.

8. Your work cites previous work on metabolic modelling. I would mention the distinction between your work and that literature more clearly: stating your contribution more plainly.

9. In the results, one wonders if neutral interactions have to have the exact same growth rate or if you have a threshold of change to call them neutral. Please clarify this.

10. For most of the introduction and results, you do not mention what the "compounds" are. To make this less abstract, saying that these compounds are proteins, sugars, fatty acids, vitamins, etc, would be useful. Also, how complex are these compounds? How different are they between the microbes in a pair?

11. Although you talk about the environment, that interactions switch based only on non-essential compounds is not very clearly stated. This could be mentioned more explicitly, as this is quite interesting.

12. When you investigate the switch between cooperation and competition, it would be helpful to try to explain one of these occurrences mechanistically. Also, it would be interesting to know if the mechanism is the same between different microbial pairs for one of these compounds leading to a switch.

13. The explanation after Equation (2) regarding the growth inequality is a little convoluted. Please simplify/clarify it.

14. In Fig. 5, the caption mentions "neutral interactions in blue", but that is not shown.

15. In Fig 4, you could break the y-axis into two sections: 1) from 0 to 30 to show obl, f comp, other and no grow, and 2) from 70 to 100 to show stay. This would enable seeing the differences in the range 0 to 30 better.

16. I find it interesting that you could separate the effect of the environment and microbial partner on the growth rate. For example, measuring the growth of microbe i when

1) all compounds needed by microbe j are added without adding microbe j itself, and discount this from the case where

2) all compounds needed by microbe j and microbe j itself are added.

17. Please deposit your code and data in a permanent repository (such as Zenodo) and consider adding a license. See, for example,

https://docs.github.com/en/repositories/archiving-a-github-repository/referencing-and-citing-content

and

https://docs.github.com/en/communities/setting-up-your-project-for-healthy-contributions/adding-a-license-to-a-repository

**Have the authors made all data and (if applicable) computational code underlying the findings in their manuscript fully available?**

Reviewer #1: Yes

Reviewer #2: **No: **The authors have not deposited their GitHub data and code in a permanent repository yet.

PLOS authors have the option to publish the peer review history of their article (what does this mean?). If published, this will include your full peer review and any attached files.

Reviewer #1: No

Reviewer #2: No

**Figure resubmission:**
---

## [Decision Letter · Decision Letter 1]

9 Jun 2025

Dear Dr. Libby,

We are pleased to inform you that your manuscript 'Competition and cooperation: the plasticity of bacterial interactions across environments' has been provisionally accepted for publication in PLOS Computational Biology.

Best regards,

Zhaolei Zhang

Section Editor

PLOS Computational Biology

Zhaolei Zhang

Section Editor

PLOS Computational Biology

Reviewer's Responses to Questions

**Comments to the Authors:**

Reviewer #1: Review of Revised Manuscript PCOMPBIOL-D-24-01869R1

The authors have substantially revised and strengthened the manuscript, addressing the previous concerns comprehensively. They now clearly define "competition" (at least one partner grows slower together than alone) and "cooperation" (both grow faster together) early on, explicitly framing their flux-balance analysis results as representing the potential for interactions, acknowledging inherent model limitations regarding realized outcomes like exploitation.

The perturbation analysis framework is significantly clarified. The authors explicitly state that trivially "redundant" (metabolically unused) compounds are excluded from their degradation analyses (Supp Fig S1). While a universal structural/functional compound similarity metric was not implemented, their pragmatic comparison of removing amino acids versus simple sugars (Supp Figs S13, S14) provides valuable functional insight, demonstrating that these classes drive distinct interaction switches and deepening the original findings.

Environmental similarity was directly addressed using the Jaccard distance metric (Supp Fig S7). The resulting analysis, showing that simple compound overlap does not cleanly separate competitive and cooperative environments, is an informative finding in itself. The ecological context is enhanced with added discussion on the potential impacts of higher-order interactions in communities and the specific role of B-vitamin dependencies, particularly relevant for obligate interactions often predicted in host-associated systems.

Statistical rigor and transparency are now robust. Figure legends detail the specific tests used (e.g., ANOVA, permutation tests with 10,000 permutations), null hypotheses, and p-values. Key results, such as the significance of transitions during environmental degradation, are supported by new analyses (Supp Fig S22). Full reproducibility is ensured through the deposition of code, statistical scripts, and sample data on GitHub and Zenodo. Visualizations (Figs 2, 3) and their legends have been improved with explanatory schematics and clearer annotations, and terminology has been standardized throughout.

In summary, the manuscript now presents a clear conceptual framework, employs rigorous and transparent methods, and offers a well-balanced discussion of its findings and limitations. The revisions have transformed it into a strong, impactful contribution.

Reviewer #2: The authors have addressed all my comments in great detail, which I thank. The updated manuscript is more precise. In particular, the expanded discussion and introduction clarify all my previous questions, and the code is available in a public repository as required by the journal. Altogether, this submission meets the requirements for publication in PLoS Computational Biology. Here are minor comments,

1. It would be good to argue more about why removing compounds is the appropriate experiment when it comes to perturbations of the environment. There are alternatives, e.g. changing the amount of the compounds. Just let us know what makes you focus on removing compounds over other possibilities.

2. Please consider adding a brief set of instructions on how to navigate/use your code. This is normally included in a *.md file in the main directory of your GitHub repository.

3. Please fix the typo in the new text of Fig. 4 " the the".

4. Please fix the broken x-axis label “… change” in Fig. 6A.

**Have the authors made all data and (if applicable) computational code underlying the findings in their manuscript fully available?**

Reviewer #1: Yes

Reviewer #2: Yes

PLOS authors have the option to publish the peer review history of their article (what does this mean?). If published, this will include your full peer review and any attached files.

Reviewer #1: No

Reviewer #2: No

---

## [Editor Report · Acceptance letter]

PCOMPBIOL-D-24-01869R1

Competition and cooperation: the plasticity of bacterial interactions across environments

Dear Dr Libby,

I am pleased to inform you that your manuscript has been formally accepted for publication in PLOS Computational Biology. Your manuscript is now with our production department and you will be notified of the publication date in due course.

With kind regards,

Lilla Horvath
